# Advanced methods for gene network identification and noise decomposition from single-cell data

Zhou Fang [1,2], Ankit Gupta [1,2], Sant Kumar [1] & Mustafa Khammash [1] ✉

Central to analyzing noisy gene expression systems is solving the Chemical Master Equation (CME), which characterizes the probability evolution of the reacting species' copy numbers. Solving CMEs for high-dimensional systems suffers from the curse of dimensionality. Here, we propose a computational method for improved scalability through a divide-and-conquer strategy that optimally decomposes the whole system into a leader system and several conditionally independent follower subsystems. The CME is solved by combining Monte Carlo estimation for the leader system with stochastic filtering procedures for the follower subsystems. We demonstrate this method with high-dimensional numerical examples and apply it to identify a yeast transcription system at the single-cell resolution, leveraging mRNA time-course experimental data. The identification results enable an accurate examination of the heterogeneity in rate parameters among isogenic cells. To validate this result, we develop a noise decomposition technique exploiting time-course data but requiring no supplementary components, e.g., dual-reporters.

Advances in modern biological technology (e.g., flow cytometry[1], time-lapse microscopy[2,3], and fluorescence proteins[4]) have substantially improved scientists' ability to investigate living biological cells. It is now well-established that the dynamics within cells are intrinsically noisy, with significant cell-to-cell heterogeneity. For example, it is known that gene expression systems typically possess a high degree of randomness due to the low molecular counts of the involved biomolecular species[5–7]. To appropriately describe such systems, stochastic continuous-time Markov chain models[8] are often employed, where each node in the chain corresponds to a specific cellular state, and the transitions between these nodes correspond to the reactions happening in the cell.

To calibrate these stochastic models, single-cell measurement data is required. One such experimental technique is Flow Cytometry which records the dynamical evolution of a heterogeneous cell population over time, providing valuable information for inferring intracellular dynamics[9] (see Fig. 1A). Time-lapse microscopy is another technology that can enable such an inference task by providing time-course data of each individual cell[10–12]. Such data is richer in

information than population-level data, as the same cells are tracked over time, preserving temporal correlations between their dynamic states. This facilitates inference of parameters, *localized* to each tracked cell[12], thereby affording a more nuanced understanding of cellular dynamics. A typical issue that arises is that only a few state variables can be dynamically tracked. However, given these measurement trajectories and a stochastic model for the dynamics, the conditional probability distribution of the unobserved state-variables can be estimated by solving what is known as the *stochastic filtering problem*[13] (see Fig. 1B). Estimation of this time-varying conditional probability distribution provides valuable insight into the dynamical behaviors of the intracellular systems and enables better feedback control design for these systems[14].

Central to stochastic modeling and analysis of biological systems is the mathematical problem of solving chemical master equations (CMEs), which consist of a collection of linear ordinary differential equations (ODEs) that determine the time-evolution of the probability distribution of the random state-vector comprising of molecular counts of all the species. The CME plays a key role in investigating the

[1]Department of Biosystems Science and Engineering, ETH Zurich, CH-4056 Basel, Switzerland. [2]These authors contributed equally: Zhou Fang, Ankit Gupta. ✉e-mail: mustafa.khammash@bsse.ethz.ch

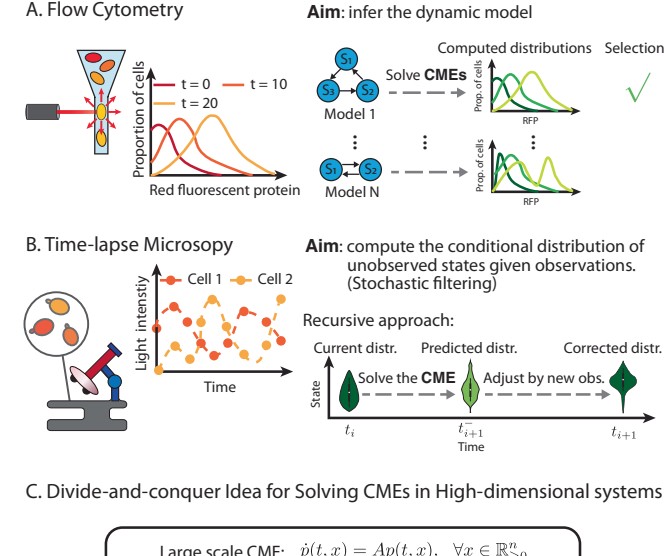

**Fig. 1 | At the heart of stochastic modeling and analysis of biological systems is the mathematical problem of solving chemical master equations (CMEs).**
**A** Inference problem associated with single-cell measurement data. To solve this inference problem, researchers need to solve CMEs to obtain single-cell distributions for many candidate models and then select the model based on the computed distributions. **B** Stochastic filtering based on single-cell time-course data. The aim is to compute the conditional probability distribution of unobserved species within a cell using the cell's time-course data. The problem is often solved in a recursive manner involving prediction steps and correction steps. The prediction steps need to solve CMEs to obtain the predicted probability distributions. **C** Our divide-and-conquer idea for solving large-scale CMEs. Our method utilizes Rao-Blackwellization and stochastic filtering (for conditional independence) to divide the original CME into several manageable sub-problems for low dimensional subsystems.

effects of noise on biological processes, such as chemotaxis[15] and cell cycle[16,17]. Solutions of the CME can also be applied to the rational design of systems, such as the genetic repressilator[18] and the antithetic integral feedback controller[19]. In these problems, scientists focus on not only the marginal distributions of individual chemical species but also their joint probability distribution, as the joint probability can unravel the interaction mechanisms between different species and their combined effects (e.g.[18,20]). In the problem of stochastic model inference from single-cell measurement data, several instances of the CME need to be solved to identify the model that best fits the data[21]. The filtering problem associated with single-cell time-course data is often solved recursively by prediction steps and correction steps[22]. The prediction steps rely heavily on solving the CME for computing predicted probability distributions[23–26].

All these applications, and many others, require efficient and accurate methods for solving CMEs. However, estimating CME solutions is very challenging, as the number of ODEs that form the CME is usually very large, and in fact in most cases of interest, it is infinite. Exact solutions of CMEs exist in very special cases[27–30], and in general one has to resort to numerical approaches to estimate CME solutions. Even though a significant amount of research effort has been devoted towards developing efficient CME solvers, the current methods are ill-equipped for solving CMEs for high-dimensional systems. This has

been a major bottleneck to the widespread adoption of stochastic models in biology.

The most commonly used methods for solving CMEs are the Monte Carlo methods and the direct approach involving state-space truncations that reduce the CME system to a finite tractable system of ODEs. Monte Carlo methods simulate multiple trajectories using algorithms such as the Gillespie algorithm[31,32], tau-leaping[33–35], or hybrid models[36,37]. The empirical distribution obtained from these simulations is used to approximate the exact probability distribution. This method can be computationally efficient, but its accuracy decreases as the dimension of the system increases, making it unsuitable for high-dimensional settings (see our discussion in the next section and Supplementary Information, Section S2.A). In contrast, direct approaches involving truncations, such as the finite state projection (FSP) approach[38–40], are very accurate, as they solve a finite-dimensional approximation of the CME directly and they provide a computable error-bound. However, since the size of the required truncated state-space scales exponentially with the system dimension, the FSP is computationally infeasible in large dimensional settings.

In some situations, CMEs for high-dimensional systems can be effectively approximated by some parametric methods. For example, when the system size is large, the linear noise approximation[41] approximates the exact probability distribution by a Gaussian distribution whose mean and variance are the parameters that need to be computed based on the underlying system. Similarly, the moment closure method closes the moment equations based on a chosen family of distributions and then tracks the first few moments[42–49]. More recently, some deep learning approaches[50–52] were established for solving CMEs thanks to the universal approximation properties of neural networks. These parametric methods are quite successful in many applications. However, their validity largely depends on the suitable choice of the family of distributions, which is not trivial to determine a priori for generic systems.

When the system is multiscale, i.e., it has reactions firing at different time scales, this curse of dimensionality can be somewhat mitigated by the method of timescale separation (e.g.[37,53–56]). In this case, a reduced-order CME system can be derived by applying the quasi-stationary assumption. Though this method is not always applicable, it is effective for analyzing many biological processes, provided they are multiscale in nature.

Dimension reduction ideas have also been applied to some hybrid methods for solving CMEs. The method of conditional moments[57] derives a reduced order CME for selected species by integrating out the moments of other species conditioned on the state of these selected ones using moment closure methods. Similarly, the uncoupled simulation method[58–60] exclusively simulates several selected species by integrating out the effect of other species using moment closure methods or entropic matching. In many biological applications, these approximation approaches (moment closure and entropic matching) can very effectively marginalize out the non-selected species, leading to enhanced computational efficiency and accuracy of these hybrid methods in estimating selected species[57–60]. However, their accuracy and reliability are not always guaranteed due to the nature of these approximations. Especially for systems where the marginalized species appear nonlinearly in the system (e.g., in the form of Hill-type kinetics), the moment dynamics involve the expectations of rational polynomials, making their accurate approximation via moment closure particularly challenging. In addition, converting approximated moments into distributions to estimate non-selected species can incur another source of error[61]. Finally, a systematic decomposition of the reacting system for optimal performance of a hybrid approach is yet to be determined.

These diverse CME-computation methods have also been extended to biological filtering problems, e.g., the particle filtering methods based on Monte Carlo simulation[25,62], the filtered FSP[63], parametric

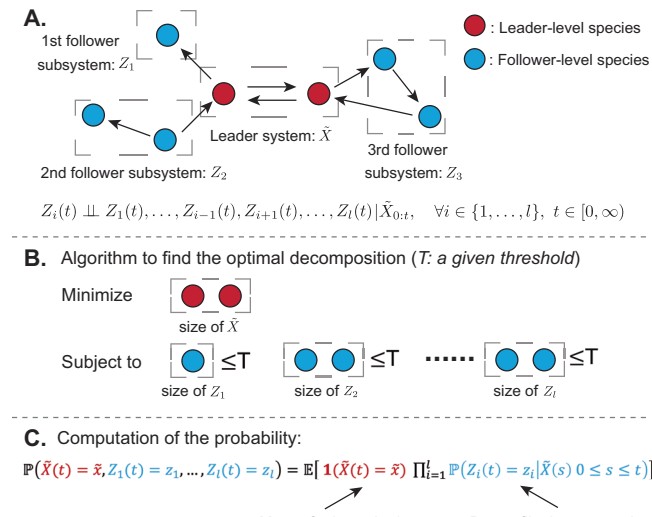

**Fig. 2 | Illustration of our approach. A** System decomposition. Our approach works by a system decomposition, which first divides a chemical reaction system into a leader system (red species) and a follower system (blue species). Then, the follower system is further decomposed into several subsystems according to some topological conditions such that the follower subsystems are conditionally independent given the trajectory of the leader system. This conditional independence is denoted by ⫫. **B** Algorithm to find the optimal decomposition. Our algorithm chooses the optimal decomposition that minimizes the size of the leader system (or maximizes the size of the whole follower system) while keeping the size of each follower subsystem below a given threshold. By this algorithm, all the subsystems are low dimensional. **C** The computation of the probability distribution. Our approach solves the CME by applying the Monte Carlo method to the leader system and a filtering approach (e.g., filtered FSP[63]) to each follower subsystem. Since our decomposition algorithm controls the size of each subsystem, our computational approach scales more favorably with the system dimension.

methods[23,24,64,65], timescale-separation based methods[26,66,67], and the hybrid method[11]. While these methods work well in many biological applications, they still exhibit the aforementioned limitations when analyzing high-dimensional systems containing more than a few species. In addition to the challenges posed by the high dimensionality of the system state, the large dimensionality of observations can cause additional challenges in the correction step of the filtering problem, rendering classical particle filtering approaches inefficient (see[13,68] for a review of particle filtering). In such situations, the block particle filter[69] and the bagged filter[70] can provide more accurate estimates through localization, albeit at the cost of introducing a small but unavoidable bias. In single-cell filtering problems, the dimension of measurements is typically low due to the limited availability of distinguishable reporters[71]. Consequently, this paper focuses on high-dimensional reaction systems with few observation channels.

In summary, there is still a paucity of methods in the current literature to effectively deal with high-dimensional stochastic reaction systems. It would, therefore, be advantageous if we could divide a high dimensional system into smaller pieces, thereby mitigating the curse of dimensionality. Motivated by this idea, we propose in this paper a modularization-based method for solving CMEs.

Our approach is inspired by a divide-and-conquer strategy enabled by probabilistic independence. Specifically, when all the species are probabilistically independent, the joint probability distribution can be derived by forming a product of all the marginal distributions. In this case, instead of directly solving the CME for the large dimensional system, we can first compute the marginal distributions by solving the reduced order CME for each species and then combine them to obtain the solution of the original CME. Suppose such a system has $n$-species with each species up to $T-1$ copies, and all

the CMEs are solved by the FSP. Then, this divide-and-conquer strategy reduces the computational complexity from $O(T^{3n})$ to $O(nT^3)$, which can result in significant improvement when the system dimension ($n$) is large. Moreover, this strategies enables parallel computation for the marginal distributions, which could further accelerate the computational process. Similar levels of complexity reduction can also be achieved with other computational methods, e.g., Monte Carlo.

Of course, the independence of species would generally not hold as the species are interacting. However, we shall adopt a new modularization strategy that exploits conditional independence between certain parts of the system, given the trajectories of intermediate species acting like conduits for inter-modular interactions. This new approach, combined with filtering (for conditional independence), makes modularization applicable to general high-dimensional networks and can significantly reduce the computational effort for solving the CME. Following this idea, our method divides the whole chemical reaction system into a leader system and several conditionally independent follower subsystems in a principled manner (see Fig. 2 for a graphic illustration). Given the decomposition, we solve the CME by employing the Rao-Blackwellization technique[72–75], which applies the Monte Carlo method to the leader system and a proper filtering approach to computing the conditional distributions of the follower subsystems (see Fig. 2C). More importantly, the system decomposition involved in our method is well-designed so that the leader system and all the follower subsystems are low dimensional (Fig. 2B). Consequently, our approach breaks down the original large scale problem into several manageable sub-problems for low-dimensional subsystems, resulting in reduced overall computational cost.

Our approach combines aspects of both the Monte Carlo method and the stochastic filtering approach, making it a hybrid method. In this sense, our optimized system-decomposition algorithm offers a way to balance the strengths of both methods for improved results. At one extreme, when all the species are treated as leader-level species, our approach reduces to the conventional Monte Carlo method. At the other extreme, when all species are treated as follower-level species, it becomes equivalent to applying the chosen filtering approach for solving the CME. In this paper, we specifically employ the filtered FSP method[63] for solving the filtering problems associated with follower subsystems. However, it is not the only choice; one can also choose other suitable filtering algorithms. Particularly, when the moment closure approach is applied for the filtering sub-problems and the leader system is simulated alone, our method is equivalent to the method in[58,59] but with a principled decomposition, which is potentially more optimal than user-defined ones.

In this paper, we demonstrate the efficacy of our approach both computationally and experimentally. First, we consider several biologically relevant in silico models and demonstrate the superior performance of our method in solving both the CMEs and the associated stochastic filtering problems. We then further develop our method and show how it can successfully leverage experimental time-course data for identifying a yeast transcription model at the single-cell resolution. This analysis illuminated the significant heterogeneities in rate parameters, even among identically cultured isogenic cells. These parameter heterogeneities can be viewed as the "extrinsic" component of the overall cell-to-cell heterogeneity[76], while the "intrinsic" component is generated by the randomness in the firing of intracellular reactions. The decomposition of total cellular homogeneity into extrinsic and intrinsic components is an active research problem[77–80], and it is fundamentally important not just for deciphering the source of variability, but also for understanding how noise affects intracellular signal processing and control[81,82]. Traditionally, the decomposition of noise in cell-to-cell heterogeneity has been conducted using experimental Flow Cytometry data in conjunction with dual reporter assays[77,79,80]. However, the synthetic implementation of these assays presents numerous challenges. These include the need to ensure the statistical

equivalence of the dual reporters and to confirm the conditional independence of the generated estimates, given the input variables.

To validate the estimates of the intrinsic and extrinsic noise from the inferred transcription models, this paper introduces a novel method for noise decomposition in gene expression from experimental data that does not rely on dual-reporter systems. This method leverages the information-richness of time-course data and capitalizes on the inherent stability of the stochastic gene-expression network. We employed this method to analyze single-cell time-course data for the yeast transcription model and compared it with the noise decomposition result generated by the inferred transcription models. The results exhibited a high degree of concordance between the two sets of estimates, thereby confirming the efficacy of our identification method.

Some terminologies are listed as follows, where the terms highlighted in bold refer to contributions in this paper.
- CME: the chemical master equation (3).
- Filtered CME: the filtering equation (5) characterizing the conditional distributions of follower subsystems.
- FSP: the finite state projection method[38].
- Filtered FSP: the FSP method for the filtered CME[63].
- PF: the particle filter, applying Monte Carlo (MC) to filtering problems.
- **RB-CME solver**: the divide-and-conquer method we propose for solving CMEs by exploiting Rao-Blackwellization and conditional independence. The method combines MC and stochastic filtering. The filtering approach is user-determined, and for the examples considered in this paper, we selected the filtered FSP.
- **RB-PF**: the Rao-Blackwellized particle filter we developed, which applies the RB-CME solver to the prediction step. Again, in the following examples, we used the filtered FSP as the chosen filtering approach within the RB-CME solver.

## Results

### Chemical master equations and the curse of dimensionality
We consider an intracellular reaction system that has $r$ reactions,

$$\nu_{1,j}S_1 + \cdots + \nu_{n,j}S_n \rightarrow \nu'_{1,j}S_1 + \cdots + \nu'_{n,j}S_n \qquad j=1,\ldots,r, \qquad (1)$$

where $S_1, \ldots, S_n$ are $n$ different chemical species, and $\nu_{i,j}$ and $\nu'_{i,j}$ are the stoichiometric coefficients. Due to the low molecular counts, this chemical reaction system is often modeled by a continuous time Markov chain[8]

$$X(t) = X(0) + \sum_{j=1}^{r} \zeta_j R_j \left( \int_0^t \lambda_j(X(s)) \mathrm{d}s \right) \qquad (2)$$

where $X(t)$ is an $n$-dimensional vector representing the molecular count of each species at time $t$, the vector $\zeta_j$ equals to $(\nu'_{1,j} - \nu_{1,j}, \ldots, \nu'_{n,j} - \nu_{n,j})^{\top}$ indicating the state change after a firing of the $j$-th reaction, $R_j(t)$ are independent unit rate Poisson processes, and $\lambda_j(\cdot)$ are the propensities indicating the rates of these reactions. In this paper, we make several mild technical assumptions for this dynamical system (see Supplementary Information, section S1) so that the process $X(t)$ is well-behaved. Under these assumptions, the probability distribution of the system (2) is characterized by the chemical master equation (CME)[8]

$$\frac{\mathrm{d}p(t,x)}{\mathrm{d}t} = \sum_{j=1}^{r} \lambda_j(x - \zeta_j)\, p(t, x - \zeta_j) - \sum_{j=1}^{r} \lambda_j(x)\, p(t,x), \qquad (3)$$

where $x \in \mathbb{Z}_{\geq 0}^n$ is the value of the state, and $p(t,x) \triangleq \mathbb{P}(X(t) = x)$ is the probability at $x$. By solving the CME, one can gain many insights into the considered biological processes.

Usually, CMEs are difficult to solve explicitly. Conventional methods to numerically solve the CME include the simulation-based Monte Carlo methods and the finite state projection (FSP) method. However, both of them scale poorly with the size of the state space and the system dimension.

A simulation-based Monte Carlo method first simulates $N$ trajectories of the system (2), denoted by $x_1(t), \ldots, x_N(t)$, and then uses the empirical distribution $p_{MC}(t,x) \triangleq \frac{1}{N} \sum_{j=1}^{N} \mathbb{1}\left(x_j(t) = x\right)$ to approximate the exact probability distribution. Here, $\mathbb{1}(\cdot)$ is the indicator function, equal to 1 when its argument is true, otherwise 0. The error of this method can be evaluated by the $L_1$ distance between $p_{MC}(t, \cdot)$ and $p(t, \cdot)$, which upper bounds the largest possible error of the Monte Carlo in estimating any particular probability. Mathematically, this error converges at the rate of $1/\sqrt{N}$, and its pre-convergence rate factor (defined by $\lim_{N \to \infty} \sqrt{N} \mathbb{E}[\|\hat{p}_{MC}(t,\cdot) - p(t,\cdot)\|_1]$) lies in a particular range shown as follows (Supplementary Information, section S2.A):

$$\begin{aligned} &\lim_{N \to \infty} \sqrt{N} \mathbb{E}\left[ \left\| \hat{p}_{MC}(t,\cdot) - p(t,\cdot) \right\|_1 \right] \\ &= \sqrt{\frac{2}{\pi}} \left[ \left( \sum_{x \in \mathbb{Z}_{\geq 0}^n} \sqrt{p(t,x)} \right) \pm 1 \right]. \end{aligned} \qquad (4)$$

This suggests that given a large sample size $N$, the error of the Monte Carlo largely depends on the value of $\sum_{x \in \mathbb{Z}_{\geq 0}^n} \sqrt{p(t,x)}$. This value tends to scale poorly with the size of the state space containing most of the probability mass. Particularly, when the probability mass is uniformly distributed on $\mathbb{S}$ states, this quantity equals $\sqrt{\mathbb{S}}$, which can be very large when $\mathbb{S}$ is big. Also, since this state size often grows exponentially with the number of species, this quantity $\sum_{x \in \mathbb{Z}_{\geq 0}^n} \sqrt{p(t,x)}$ tends to scale poorly with the system dimension $n$. This point can be clearly seen when the molecular counts of different chemical species are independent; in this case, $\sum_{x \in \mathbb{Z}_{\geq 0}^n} \sqrt{p(t,x)}$ does grow exponentially with $n$, as its value is equal to $\prod_{i=1}^{n} \left( \sum_{x_i \in \mathbb{Z}_{\geq 0}} \sqrt{p_i(t,x_i)} \right)$ with $p_i(t, \cdot)$ the marginal distribution for the $i$-th species. In summary, the error of a Monte Carlo method tends to scale unfavorably with the size of state space and system dimension, and, thus, this method usually performs poorly in high-dimensional problems.

In contrast, the FSP approach[38], which directly solves a truncated CME on a large but finite state space, is very accurate for arbitrary reaction systems. However, the computational complexity of this method scales cubically with the size of the truncated state space. Moreover, since the size of the finite state space scales exponentially with $n$, the FSP is computationally demanding for high-dimensional problems.

In conclusion, both Monte Carlo and FSP methods suffer the curse of dimensionality.

### Motivating example: beat the curse of dimensionality through probabilistic independence
Despite the challenges mentioned above, the CME can be efficiently solved for large dimensional systems when all the states in $X(t)$ are probabilistically independent. In such cases, the probability distribution can be obtained by combining all the marginal distributions. Therefore, the large-scale CME can be divided into several manageable sub-problems, each of which only solves a CME for obtaining one marginal distribution.

To further elaborate this divide-and-conquer approach, we consider a toy example where all the species are independent, each with no more than $\mathbb{S} - 1$ copies, and their marginal probability mass is uniformly distributed in the state space. When straightforwardly applying the FSP to such a system, we encounter a computational complexity of $O(\mathbb{S}^{3n})$, as discussed in the previous section. In contrast, the divide-and-conquer strategy reduces the complexity to $O(n\mathbb{S}^3)$, which is much more favorable when both $n$ and $\mathbb{S}$ are large. In addition to this complexity reduction, this divide-and-conquer strategy also enables parallel computation for the marginal distributions, which could

further reduce the computational time. Such an improvement also occurs when the Monte Carlo method is employed. According to (4), the classical Monte Carlo method needs to generate $\frac{s^n}{\epsilon^2}$ samples to reach an accuracy of $\epsilon$. In contrast, each marginal distribution only needs $\frac{s}{\epsilon^2}$ samples to reach this accuracy. Since the errors in estimating these marginal distributions contribute additively to the error in the joint distribution estimate (Supplementary Information, section S2.A), the divide-and-conquer strategy only needs $\frac{n^3 s}{\epsilon^2}$ samples to attain the accuracy of $\epsilon$. This figure is significantly less than the sample size ($\frac{s^n}{\epsilon^2}$) required by the conventional method. Overall, this modularization method powered by probabilistic independence can significantly reduce the required computational complexity and beat the curse of dimensionality.

Nevertheless, species are generally not probabilistically independent as they are interacting typically. In what follows, we propose a new modularization approach exploiting conditional independence together with Rao-Blackwellization. This new strategy, combined with filtering (for conditional independence), makes modularization applicable to general high-dimensional networks and can also significantly reduce the computational effort for solving CMEs.

## Modularization-based Rao-Blackwell method for solving CMEs

We propose a modularization method for solving CMEs by exploiting Rao-Blackwellization and conditional independence. Specifically, our method first uses an automated algorithm to decompose the system into two parts: $\tilde{X}(t)$ (denoted as the leader system) and $Z(t)$ (termed as the follower system) (see Fig. 2). The detailed decomposition strategy will be discussed later in this section. Based on this leader-follower decomposition, we also divide reaction vectors $\zeta_j$ into $\zeta_j^{\tilde{X}}$ and $\zeta_j^Z$, where $\zeta_j^{\tilde{X}}$ indicates the state change of the leader system, and $\zeta_j^Z$ indicates the state change of the follower system. Moreover, we further decompose the follower system $Z(t)$ into several lower-dimensional subsystems, $Z_1(t), ..., Z_l(t)$, such that the following topological conditions are satisfied.

C1 Each reaction in (1) involves a maximum of one follower subsystem (meaning that at most one follower subsystem can influence this reaction's propensity or have its state altered by this reaction).

C2 The reactions with the same non-zero $\zeta_j^{\tilde{X}}$ involve a maximum of one follower subsystem (meaning that at most one follower subsystem can influence the propensities of these reactions or have its state altered by these reactions).

For ease of notations, we rearrange the order of species such that $X(t) = \left(\tilde{X}(t), Z_1(t), ..., Z_l(t)\right)$; also, for every state $x$, we write $x = (\tilde{x}, z_1, ..., z_l)$, where $\tilde{x}$ is the state of the leader system, and $z_i$ ($i = 1, ..., l$) is the state of the follower system. Under the conditions above, the follower subsystems are conditionally independent given the trajectory of the leader system (see the proof in Supplementary Information, Section S3.A). Moreover, the conditional probability distribution $\pi_{Z_i|\tilde{X}}(t, z_i) \triangleq \mathbb{P}\left(Z_i(t) = z_i | \tilde{X}(s), 0 \le s \le t\right)$ is characterized by a set of differential equations with jumps (heuristically derived in[25,58–60] and rigorously verified in ref. [63]):

$$
\begin{aligned}
d\pi_{Z_i|\tilde{X}}(t, z_i) =\; & f_1\left(\tilde{X}(t), z_i, \pi_{Z_i|\tilde{X}}(t, \cdot)\right) dt + f_2\left(\tilde{X}(t), z_i, \pi_{Z_i|\tilde{X}}(t, \cdot)\right) dt \\
& + \sum_{j=1}^r \mathbb{1}\left(\tilde{X}(t) - \tilde{X}(t^-) = \zeta_j^{\tilde{X}}\right) g_j\left(\tilde{X}(t^-), z_i, \pi_{Z_i|\tilde{X}}(t^-, \cdot)\right)
\end{aligned}
\tag{5}
$$

whose detailed expression is given in Supplementary Information, Section S3.A. Here, $f_1$ represents the prediction of the conditional distribution based on the dynamical model, and the remaining terms correspond to the corrections to the estimates in accordance with the dynamics of the observable species. In this paper, we call (5) the filtered CME because it computes the conditional distribution, which is a solution of a filtering problem. A schematic illustration of the decomposition introduced above is presented in Fig. 2A. Here, we

should note that this conditional independence holds only when the entire trajectory of the leader species is given up to the current time $t$. If only the current state of the leader species is provided, this conditional independence may not necessarily hold.

By this decomposition and the law of total expectation, we can rewrite the probability distribution by $p(t, \tilde{x}, z_1, ..., z_l) = \mathbb{E}\left[\mathbb{1}\left(\tilde{X}(t) = \tilde{x}\right) \prod_{i=1}^l \pi_{Z_i|\tilde{X}}(z_i)\right]$. Based on it, we design an Rao-Blackwell method for CMEs, which applies Monte Carlo to the leader system and a filtering approach to each follower subsystems. Concretely, we first generate $N$ simulations of the system (2) and denote their leader-system parts by $\tilde{x}_1(t), ..., \tilde{x}_N(t)$, respectively. Then, for each simulated trajectory and each follower subsystem, we calculate the conditional probability distribution $q_j^i(t, z_i) \triangleq \mathbb{P}\left(Z_i(t) = z_i | \tilde{X}(s) = \tilde{x}_j(s), 0 \le s \le t\right)$ using a stochastic filtering approach. One has the flexibility to choose suitable filtering approaches (e.g., Monte Carlo method[25], filtered FSP[63] etc.) for computing these conditional distributions. In the examples considered in this paper, we specifically employ the filtered FSP[63], which solves (5) directly on a large but finite state space. Finally, the exact probability distribution is approximated by the quantity

$$
\hat{p}_{RB}(t, x) = \frac{1}{N} \sum_{j=1}^N \left[\mathbb{1}\left(\tilde{x}_j(t) = \tilde{x}_j\right) \prod_{i=1}^l q_j^i(t, z)\right].
\tag{6}
$$

We name this algorithm the Rao-Blackwellized CME solver (RB-CME solver). Mainly, this Rao-Blackwellized method transforms the original problem of solving the CME into several potentially low dimensional subproblems. Therefore, our method tends to scale more favorably with the system dimension.

The RB-CME solver can be seen as a principled way to combine the Monte Carlo method and the chosen filtering approach. Particularly, when all the species are classified as leader-level species, this method becomes a Monte Carlo method. When all the species are classified as follower-level species, the filtered CME becomes the CME, and the RB-CME solver is equivalent to applying the chosen filtering approach to the CME with the scenario of non-existent observation. In the scenarios between these extreme cases, our method is a combination of the two approaches, capable of leveraging their advantages to achieve better performance than either method in isolation. When the filtered FSP is applied to the follower subsystems, our method is computationally more tractable than the original FSP approach in high dimensional cases, as our method applies the filtered FSP to each follower subsystem separately.

Given the same sample size $N$, the RB-CME solver is no less accurate than the conventional Monte Carlo method, if all the conditional distributions $q_j^i(t, z_i)$ are computed precisely. Basically, the first layer of the RB-CME solver is a Monte Carlo approach; therefore, its $L_1$ error has a convergence rate $\sqrt{N}$ and its pre-convergence rate factor depends on the variance of the random variable it generates. The expression of this pre-convergence factor is given in Supplementary Information, Section S3.C. Note that the variance of $\mathbb{1}\left(\tilde{X}(t) = \tilde{x}\right) \prod_{i=1}^l \pi_{Z_i|\tilde{X}}(z_i)$ (used for the RB-CME solver) is no greater than that of $\mathbb{1}(X(t) = x)$ (used for the conventional Monte Carlo method) due to the law of total variance. So, we can conclude the superior performance of our method in the sense of the pre-convergence rate factor, i.e., (see Supplementary Information, Section S3.C)

$$
\begin{aligned}
& \lim_{N \to \infty} \sqrt{N} \mathbb{E}\left[\left\|\hat{p}_{RB}(t, \cdot) - p(t, \cdot)\right\|_1\right] \\
& \le \lim_{N \to \infty} \sqrt{N} \mathbb{E}\left[\left\|\hat{p}_{MC}(t, \cdot) - p(t, \cdot)\right\|_1\right].
\end{aligned}
$$

Particularly, if the conditional probability distribution $\pi_{Z_i|\tilde{X}}(z_i)$ only depends on the final state of the leader system $\tilde{X}(t)$ and is independent

of any of its historical information, then the RB-CME solver is equivalent to the time scale separation method, which eliminates all the dynamics of the follower-level species using quasi-stationary assumption. In this case, our method's error is equal to the Monte Carlo method's error for estimating $\tilde{X}(t)$ solely (Supplementary Information, Section S3.C), and its pre-convergence rate factor is far less than that in (4) when the dimension of $\tilde{X}(t)$ is much less than that of $X(t)$. A more comprehensive discussion about the connections between the time-scale separation method and our approach is provided in Supplementary Information, Section S3.E. These facts indicate that the RB-CME solvers can be far more accurate than the Monte Carlo method for general high-dimensional problems.

Despite this error reduction, given the same sample size $N$, the RB-CME solver is more time-consuming and memory-demanding than the Monte Carlo method, as our method needs to additionally compute and store the marginal distributions $q_j^i(t, z_i)$. Then, the natural question arising is whether our method has superior performance under the same time cost or at the same accuracy level. This poses a challenging mathematical problem. Specifically, this requires an exact computation of the variance of $\mathbb{1}\left(\tilde{X}(t) = \hat{x}\right) \prod_{i=1}^{l} \pi_{Z_i|\tilde{X}}(z_i)$, which is case-dependent and difficult to obtain in a general explicit form. Instead of pursuing a universal solution to this problem, we aim to obtain insights through a case study of the motivating system considered in the preceding section. Recall that for this $n$-dimensional system, the Monte Carlo method needs $\frac{S^n}{\epsilon^2}$ samples (equivalent to a complexity of $O(\frac{S^n}{\epsilon^2})$) to reach an accuracy of $\epsilon$, where $S$ is the state space size for each species. Now, we apply our RB-CME solver to this problem by classifying $n_1$ species as leader species and the remaining as follower-level species. We also assume that these follower-level species can be divided into $\frac{n - n_1}{n_2}$ follower subsystems, each containing $n_2$ species, and the error of the RB-CME solver equals the error in estimating the leader species. In this secnario, the RB-CME solver needs $\frac{S^{n_1}}{\epsilon^2}$ samples to reach an accuracy of $\epsilon$. Moreover, when we apply the filtered FSP to compute the filtered CME for each follower subsystem, the RB-CME solver has a complexity of $O(\frac{S^{n_1}}{\epsilon^2} \times \frac{n - n_1}{n_2} S^{3n_2}) = O(\frac{n - n_1}{n_2 \epsilon^2} S^{n_1 + 3n_2})$. We can observe that this complexity is much reduced compared with that of the Monte Carlo method $O\left(\frac{S^n}{\epsilon^2}\right)$, when the system dimension $n$ is large, and $n_1$ and $n_2$ are small. In conclusion, the RB-CME solver has better performance if the leader system and all the follower subsystems have small state spaces.

As demonstrated in the above example, system decomposition has an important role in determining the performance of the RB-CME solver. Here, we present a principled method for system decomposition aimed at maximizing the efficiency of our approach. For each follower subsystem, we define $SS_i$ as the size of its truncated state space containing the most probability mass. Intuitively, the RB-CME solver is more accurate and efficient when the leader system and all the follower subsystems have small sizes in (truncated) state spaces. Mathematically, this suggests that $\prod_{i=1}^{l} SS_i$ (the size of the whole follower system) should be large, while $\max_i SS_i$ (the size of the largest follower subsystem) should be small. In most cases, these two optimization objectives cannot not be achieved simultaneously. Consequently, we choose a leader-follower decomposition (among all) that maximizes the size of the whole follower system ($\prod_{i=1}^{l} SS_i$) while keeping the size of each individual follower subsystem ($SS_i$) below a given threshold. Even though this optimization problem is non-convex, we can still tackle it by exhaustive search due to the discrete nature of the problem. The detailed algorithm following this strategy is provided in Supplementary Information, Section S3.D. With this decomposition algorithm, we effectively divide the problem of solving CMEs into several lower-scale computational sub-problems, and, consequently, our approach scales more favorably with the system dimension in terms of accuracy and efficiency.

Finally, we summarize the procedure of the RB-CME solver as follows (also see a schematic illustration in Fig. 2).

1. Decompose the system into a leader system and several follower subsystems using the algorithm in Supplementary Information, Section S3.D.
2. Generate N simulations of the whole system and keep only the leader system trajectories $\tilde{x}_1(t), \dots, \tilde{x}_N(t)$.
3. For each trajectory $\tilde{x}_j(\cdot)$ and each subsystem $Z_i(t)$, solve $q_j^i(t, z_i) \triangleq \mathbb{P}\left(Z_i(t) = z_i | \tilde{X}(s) = \tilde{x}_j(s), 0 \leq s \leq t\right)$ by applying the filtered FSP to the filtered CME (5).
4. Compute the result by (6).

**Rao-Blackwell method for solving the filtering problem**

With modern time-lapse microscopes, scientists can measure the time trajectory of some intracellular species (e.g., fluorescent reporters) and use these measurements to infer the dynamical states of unobserved species (e.g., the gene state). This process, known as stochastic filtering for intracellular reaction systems, allows scientists to gain insights into unobserved chemical species. This in turn can lead to the development of improved control strategies for the reacting process[14]. Here, we apply the proposed RB-CME solver to this filtering problem.

Mathematically, we can model the observation channels by

$$Y(t_i) = h(X(t_i)) + \Sigma W_i \tag{7}$$

where $t_i$ are the observation time points, $Y(t_i)$ is a vector of observations with each element corresponding to a particular light frequency, $h(\cdot)$ is a vector-valued function indicating the ideal relation between the measurement and the system state, $W_i$ are vectors of independent standard Gaussian noise, and $\Sigma$ is a diagonal matrix indicating observation noise intensities. When the observations $Y(t_i)$ are one dimensional, we denote the $1 \times 1$ matrix $\Sigma$ as $\sigma$. Stochastic filtering aims to compute the conditional probability distribution of the state $X(t_i)$ (for every $i = 1, 2, \dots$) given the observations up to time $t_i$, i.e., $\pi_{t_i}(x) \triangleq \mathbb{P}(X(t_i) = x | Y(t_s), 1 \leq s \leq i)$. Let us denote $\rho_{t_i}(x) \triangleq \mathbb{P}(X(t_{i+1}) = x | Y(t_s), 1 \leq s \leq i)$. By Bayes' rule, $\pi_{t_i}(x)$ satisfies the following recursive formulas[22]

$$\rho_{t_{i+1}}(x) = \sum_{x' \in \mathbb{Z}_{\geq 0}^n} \mathbb{P}\left(X(t_{i+1}) = x | X(t_i) = x'\right) \pi_{t_i}(x') \tag{8}$$

$$\pi_{t_{i+1}}(x) \propto L\left(Y(t_{i+1}) | x\right) \rho_{t_{i+1}}(x) \tag{9}$$

where $L(y|x)$ is the density function of the distribution $\mathbb{P}(Y(t_{i+1}) \in \mathrm{d}y | X(t_{i+1}) = x)$ (usually called the likelihood function). We can interpret (8) as the prediction of the state at the next time point $t_{i+1}$ using the observation up to the current time $t_i$. We interpret (9) as the adjustment of the prediction according to the next observation. Note that the prediction step (8) is actually solving a CME with $\pi_{t_i}(\cdot)$ being the initial probability distribution and $\rho_{t_{i+1}}(\cdot)$ being the final solution. Consequently, the filtering problem can be seen as a combination of the usual CME and an adjustment step.

Following the idea above, one can solve the single-cell filtering problem by applying various CME solvers to the prediction step. Conventional methods include the particle filter that uses the Monte Carlo method for (8) (see refs. 13,68 for a review of particle filtering and [25,26,66,67] for its application to single-cell data) and the direct approach that uses the FSP for (8). Similar to the situation in solving CMEs, these two approaches scale poorly with the system dimension when solving the filtering problem. Specifically, the particle filter has a similar $L_1$ error to the Monte Carlo method (for solving CMEs) when they are applied to the same system (Supplementary Information, Section S2.B), and, therefore, the particle filter can be inaccurate when the system dimension is large. Besides, applying the FSP to the filtering problem can be time-consuming because the size of the state space scales exponentially with $n$.

Here, we solve the filtering problem by applying the RB-CME solver to (8); we call this approach the Rao-Blackwellized particle filter (RB-PF). In this case, the follower subsystems need to be conditionally independent given the trajectories of both the leader system and the observation. To this end, we require the following condition for the system decomposition in addition to C1 and C2. (The proof is given in Supplementary Information, Section S4.A.)

C3 Each observation channel cannot be affected by more than one follower subsystem. In other words, each component of $h(\cdot)$ can depend on one follower subsystem at most (besides the leader system).

This requirement automatically holds in many practical applications where fluorescent reporters are used as probes. In these cases, experimentalists usually use different fluorescent reporters to tag different genes, thereby establishing a one-by-one correspondence between observed signals and actual gene products. We provide the modified leader-follower decomposition algorithm for the filtering problem in Supplementary Information, Section S4.A, and the detailed algorithm of the RB-PF in Supplementary Information, Section S4.B.

We found that when applied to the same chemical reaction system, the RB-PF (for the filtering problem) has similar accuracy to the RB-CME solver (for solving CMEs) under certain regularity conditions (Supplementary Information, Section S4.C), which suggests that the RB-PF also scales favorably with the system dimension. Also, this means that for a given reaction system, if the RB-CME solver can accurately solve its CME, then the RB-PF can also accurately solve its filtering problem and vice versa.

## Rao-Blackwell method for cell-specific parameter identification

The biological dynamics occurring within cells are not exactly known. This gives rise to another important topic in biology, i.e., parameter identification, which aims to calibrate mathematical models for biological processes from given datasets. Securing a good model can provide a deep understanding of the underlying biological mechanisms and allow for more accurate predictions of system behaviors. In this section, we present a method for the identification of cell-specific models by exploiting the RB-CME solver.

In the identification problem, we consider that each cell can undergo $r$ reactions as in (1), consisting of all possible biological mechanisms within the cell. Also, the system follows a dynamical equation $X(t) = X(0) + \sum_{j=1}^{r} \zeta_j R_j\left(\int_0^t \lambda_j(\Theta, X(s)) ds\right)$, which is similar to (2) with the exception that the propensity functions are also dependent on unknown parameters $\Theta \in \mathbb{R}^{\bar{r}}$ (e.g., reaction constants and hill coefficients). Usually, these parameters can take any value within certain ranges; however, for simplicity, we consider that $\Theta$ only takes values in a discrete set $\mathbf{\Theta}$ that provides a fine-grained representation of the continuous parameter region. As in the filtering problem, we consider that a cell is tracked and measured under a microscope at different time points $(t_1, \ldots, t_{n_f})$, with the observations being represented by $Y(t_1), \ldots, Y(t_{n_f})$. Ultimately, this cell-specific identification problem aims to calculate the conditional probability of the parameters $\Theta$ given the measurements, i.e., $P\left(\Theta = \theta | Y(t_s), 1 \le s \le n_f\right)$.

Similar to stochastic filtering, this identification problem can also be solved in a recursive manner. Essentially, the parameters $\Theta$ can be viewed as the state of some additional special chemical species in the system, which can take non-integer values and remain constant over time. From this perspective, this cell-specific identification problem aims to infer the hidden states of these special species, a task closely aligned with the filtering problem introduced in the preceding section. Therefore, by denoting the condition distributions $\pi_{t_i}(\theta, x) \triangleq \mathbb{P}(\Theta = \theta, X(t_i) = x | Y(t_s), 1 \le s \le i)$ and

$\rho_{t_{i+1}}(\theta, x) \triangleq \mathbb{P}(\Theta = \theta, X(t_{i+1}) = x | Y(t_s), 1 \le s \le i)$, we can solve this identification problem using the following recursive formulas:

$$\rho_{t_{i+1}}(\theta, x) = \sum_{x' \in \mathbb{Z}_{\ge 0}^n} \mathbb{P}(X(t_{i+1}) = x | \Theta = \theta, X(t_i) = x') \pi_{t_i}(\theta, x') \tag{10}$$

$$\pi_{t_{i+1}}(\theta, x) \propto L(Y(t_{i+1})|x) \rho_{t_{i+1}}(\theta, x) \tag{11}$$

$$\mathbb{P}\left(\Theta = \theta | Y(t_s), 1 \le s \le n_f\right) = \sum_{x \in \mathbb{Z}_{\ge 0}^n} \pi_{t_{n_f}}(\theta, x). \tag{12}$$

Here, (10) represents the prediction of the entire state $(\Theta, X(t))$ at the subsequent time point $t_{i+1}$ using the observation up to the current time $t_i$. Meanwhile, (11) adjusts this prediction using the next measurement at time $t_{i+1}$. Finally, (12) obtains the result of parameter identification by marginalization. Since $\Theta$ can be viewed as the state of additional chemical species, the probability distribution of $(\Theta, X(t))$ evolves according to an augmented CME (Supplementary Information, Section S5.B), a minor extension of the classical CME. Consequently, the prediction step (10) corresponds to solving this augmented CME with $\pi_{t_i}(\cdot)$ as the initial probability distribution and $\rho_{t_{i+1}}(\cdot)$ as the solution at the final time. Thus, this parameter identification problem can also be interpreted as a combination of solving the augmented CMEs and making subsequent adjustments as new measurements arrive.

From these viewpoints, we propose using the RB-PF to solve this identification problem, i.e., applying the RB-CME solver to the prediction step, which involves solving the augmented CME. To achieve this, we need to classify all the chemical species and parameters into leader and follower systems, and the follower subsystems must satisfy C1–C3 so that they are conditionally independent given the trajectories of both the leader system and the measurement. In addition, it is noteworthy that classical particle filtering is generally ineffective for the inference of static hidden state variables (e.g., model parameters) due to sample degeneracy[83]. Several improved methods, e.g., the resample-move method[84,85], regularized particle filter[67,83,86] and nested filters[87,88], address this issue by introducing artificial noise to the static variables. However, selecting the appropriate artificial noise intensity to balance mitigating sample degeneracy against minimizing additional bias is also very challenging, with few theoretical results on how this can be simultaneously achieved. For these reasons, we require our identification algorithm to classify all the parameters as follower components (C4), which avoids the aforementioned problems by allowing their inference to be aided by a particle-free approach (e.g., filtered FSP) rather than being identified purely by classical particle filtering.

C4 All the model parameters $\Theta$ are classified as follower components of the system.

The detailed algorithm for the leader-follower decomposition adhering to C1–C4 is provided in Supplementary Information, Section S5.C.1, and the Rao-Blackwell algorithm for parameter identification is presented in Supplementary Information, Section S5.C.2.

The concept of the RB-PF has long been introduced for filtering problems in general state-space models[74,75]. However, its adaptation to parameter identification is relatively unexplored, even though some contributions in this direction exist. A pioneering work[11] proposed a method that explicitly marginalizes out the uncertainty of parameters given the dynamics of the chemical species and shows strong performance with both numerical and experimental data. Several significant differences exist between our approach and that one. First, the method in[11] is tailored specifically for systems where each propensity depends linearly on one associated parameter. This constraint ensures the Gamma conditional distributions of the parameters, but it fails in many

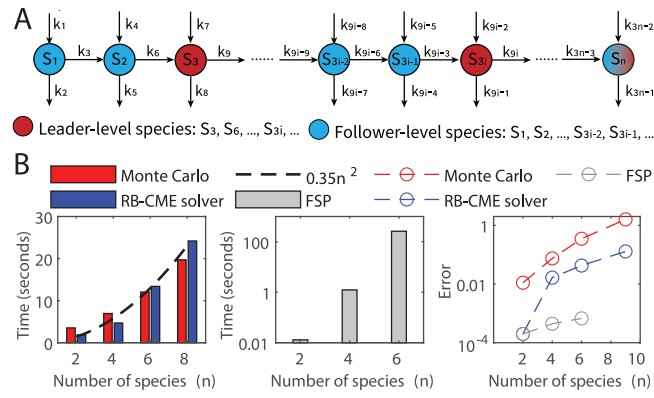

**Fig. 3 | Scalability of the RB-CME solver in a class of linear networks. A** A class of linear networks that consists of three types of reactions: the production, degradation, and conversion of $S_i$ into $S_{i+1}$ ($i = 1, ..., n-1$). All the reactions follow mass-action kinetics, and their reaction constants are $k_1 = 2.4$, $k_{3n-1} = 1.6$, $k_{3i+1} = 0.9$, $k_{3i-1} = 0.6$, and $k_{3i} = 1$ ($i = 1, ..., n-1$). At the initial time, each species has a Poisson probability with mean 0.5, and all of them are independent. **B** The scalability of the Monte Carlo method (with $10^5$ samples), RB-CME solver (with $10^4$ samples), and finite state projection approach (with the truncated space $\bigotimes_{i=1}^{n}\{0,1,...,9\}$) when solving the CME at time 10. We used the filtered FSP as our chosen filtering approach in the RB-CME solver. In the third block, the error is evaluated by the $L_1$ distance between the numerical solution and the exact solution of the CME. This panel tells that the computational time of the Monte Carlo method and that of the RB-CME solver both grow quadratically with the system dimension ($n$), whereas the computational time of the FSP method grows exponentially with $n$. Moreover, the error of the Monte Carlo method and that of the RB-CME solver both grow exponentially with $n$, but the latter grows much slower than the former (see the slopes of the linear-like curves in the log-domain). Notably, the RB-CME solver is as accurate as the FSP method when $n = 2$. It is because, in this case, no leader-level species exist, and, therefore, the RB-CME solver with the filtered FSP as the chosen filtering approach is equivalent to the FSP method. Source data are provided as a Source Data file.

cases, e.g., when parameters jointly and nonlinearly affect propensities (e.g., Michaelis-Menten kinetics and Hill-type dynamics). In contrast, our approach is not constrained by the type of kinetics and therefore has broader applicability. Second[11], relies on classical particle filtering for the inference of all chemical species, whereas our approach can classify some species as follower components, which allows us to infer them with the assistance of a suitable filtering approach (e.g., the filtered FSP). To conclude, though both approaches draw inspiration from the Rao-Blackwellization technique, they are quite different.

### Setup of numerical case studies

Next, we illustrate our approach for solving CMEs and stochastic filtering problems through several biologically relevant numerical examples. Unless stated otherwise, all experiments were performed on the Euler computing cluster at ETH Zurich, utilizing computational nodes with 2.25-GHz, 12-core CPUs. The code is available on GitHub: "https://github.com/ZhouFang92/Rao-Blackwellized-CME-solver".

### Application of the RB-CME solver to a class of linear reaction systems

To demonstrate the scalability of the RB-CME solver, we applied our approach to a class of expandable linear networks shown in Fig. 3A and compared its performance with the traditional approaches. Specifically, the linear network consists of $n$ chemical species and three types of reactions: the production ($\emptyset \to S_i$), the degradation ($S_i \to \emptyset$), and the conversion of $S_i$ into $S_{i+1}$ ($i \le n-1$). We modeled the reactions to follow mass-action kinetics with the rate constants presented in the caption of Fig. 3. At the initial time, the molecular counts of different species are independent and have a Poisson probability with mean 0.5. In this setting, the associated CME can be solved by a multivariate

Poisson distribution whose mean evolves according to the deterministic dynamics of the system[27].

We first compared the scalability of different approaches with respect to accuracy and efficiency. For the FSP method, we truncated the state space by $\bigotimes_{i=1}^{n}\{0,1,...,9\}$, which contains most of the probability. Similarly, with the RB-CME solver, we truncated the state space for each follower-level species by $\{0,1,...,9\}$ and required each follower subsystem to contain no more than 100 states. In this setting, our leader-follower decomposition algorithm consistently classifies species $S_{3i}$ ($i = 1, 2, ...$) as leader-level species and the rest as follower-level species. Each pair of $S_{3i-2}$ and $S_{3i-1}$ ($i = 1, 2, ...$) form a follower subsystem. In this example, we selected the filtered FSP as the filtering approach to be used in the RB-CME solver. Also, we set the RB-CME solver and the Monte Carlo method to have respectively $10^4$ samples and $10^5$ samples so that their computational time is relatively the same. The experimental results are shown in Fig. 3B.

Figure 3B indicates that the computational time of the RB-CME solver scales well with the system dimension ($n$), and its accuracy is much better than that of the Monte Carlo method at the same time cost. Specifically, the first block in Fig. 3B shows that the computational time of the Monte Carlo method grows quadratically with $n$, which is because the Gillespie method has the computational complexity $\mathcal{O}(\texttt{\#reaction channels} \times \texttt{\#reaction firing events})$ (see[89, Section III]), and both of these quantities grow linearly with $n$ in this example. For the FSP method, the algorithm's time-complexity is linear with the truncated space size, which scales exponentially with the dimension $n$ in this example, so its cost also grows exponentially with $n$ (see the second block of Fig. 3B). In contrast, though the RB-CME solver utilizes the filtered FSP to the follower system whose size also grows exponentially with $n$, its computational time still scales quadratically with $n$ (see the first block of Fig. 3B). This reduced computational complexity is because we apply the filtering approach to each follower subsystems separately rather than the whole system. Therefore, in our algorithm, the computational cost of the FSP part becomes $\mathcal{O}(\texttt{size of the largest follower subsystem} \times \texttt{\#follower subsystems})$ where the first term is fixed ($= 100$), and the second term scales linearly with $n$. Additionally, parallel computing further aids in reducing the computational time. As for the accuracy, the third block of Fig. 3B tells that the error of the Monte Carlo method and that of the RB-CME solver both scale exponentially with $n$, but the latter grows much slower than the former. Notably, when the system dimension is two, i.e., all the species are follower-level species, the RB-CME solver (with the filtered FSP as the chosen filtering approach) is equivalent to the FSP method, and both approaches are equivalently accurate. All these results indicate that the RB-CME solver is a good compromise between the Monte Carlo method and the FSP approach, and it is more favorable for high-dimensional problems.

To further understand the benefit of the RB-CME solver, we investigated its use in more detail on the linear network with six species (depicted in Fig. 4A). From the results, we can observe that both the Monte Carlo method and the RB-CME solver converge to the exact probability distribution at the rate of $1/\sqrt{N}$, which agrees with the law of large numbers (see Fig. 4B). Moreover, given the same sample size (resp., the same computational time), the RB-CME solver is significantly more accurate than the Monte Carlo method with an improvement of 20 times (resp., 8 times), respectively. We also studied the performance of both approaches in estimating the marginal distributions when the time costs are relatively the same (see Fig. 4C). The result shows that both methods accurately approximate the marginal distribution for individual species, but their performance in estimating joint probabilities is very different. Concretely, the RB-CME solver is more accurate in estimating the follower system, especially the first follower subsystem consisting of $S_1$ and $S_2$, but it is less accurate in estimating the leader systems (see the last block in Fig. 4B). The relative inaccuracy of the RB-CME solver for the leader system is attributed

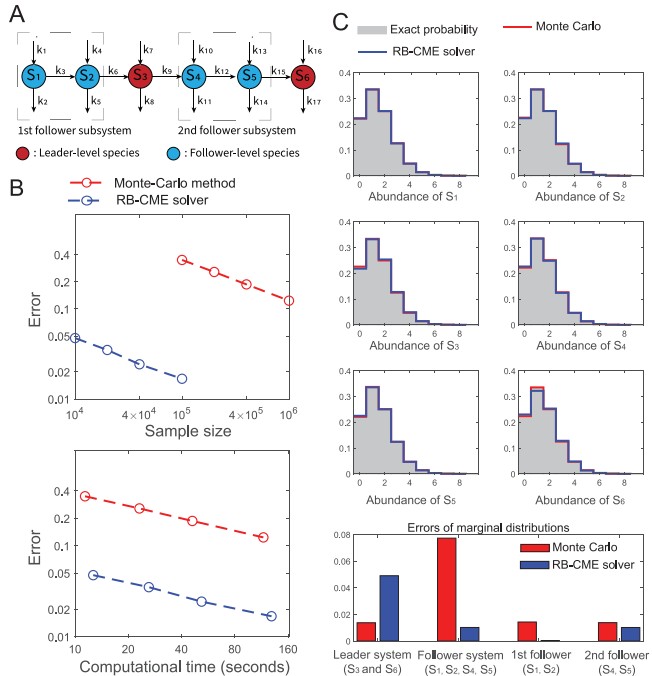

**Fig. 4 | Comparison of the RB-CME solver and Monte Carlo method in the linear network with six species. A** The diagram of the linear network with six species: all the settings are the same as that in Fig. 3. **B** Convergence of both approaches in terms of the $L_1$ error. Both methods converge at the rate of $1/\sqrt{N}$ or $1/\sqrt{T}$, where $N$ and $T$ are the sample size and computational time, respectively. This also implies that for both approaches, the computational time is proportional to the sample size (see the last block). With the same sample size, the RB-CME solver is 20 times more accurate than the Monte Carlo method, and at the same time cost, the RB-CME solver is 8 times more accurate. **C** Performance of the RB-CME solver (with $10^4$ samples) and Monte Carlo method (with $10^5$ samples) in estimating the marginal distributions. Both methods accurately approximate the marginal distributions for individual species, but they perform quite differently in estimating the joint probabilities. Specifically, the RB-CME solver is more accurate in estimating the follower system, particularly the first follower subsystem consisting of $S_1$ and $S_2$, but it is less accurate in estimating the leader system. Source data are provided as a Source Data file.

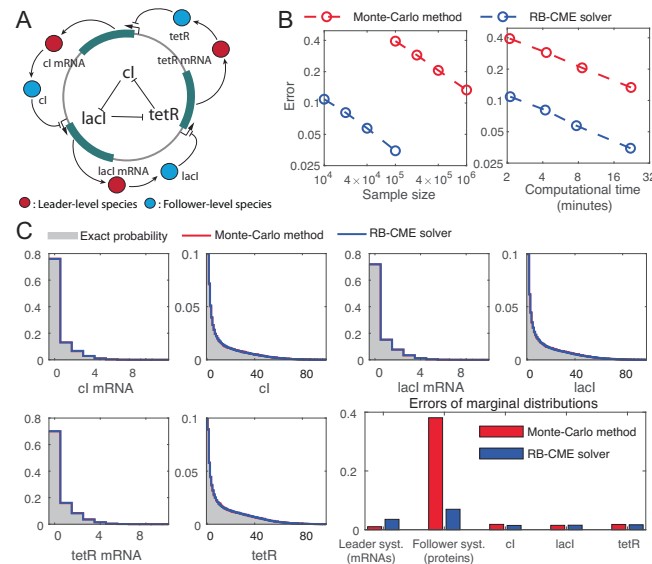

**Fig. 5 | Performance of the RB-CME solver for the repressilator. A** Diagram of the repressilator with three gene expression systems whose proteins cyclically repress each other. Our method classifies all the mRNAs as leader-level species and all the proteins as follower-level species. **B** Convergence of the Monte Carlo method and the RB-CME solver (with the filtered FSP as the chosen filtering approach). We depict the error by the sum of the $L_1$ errors in estimating the leader system and the follower system. The exact probability distribution is approximated by the Monte Carlo method with $3 \times 10^9$ samples. Given the same sample size or the same time cost, the RB-CME solver is much more accurate than the Monte Carlo method. **C** Performance of the RB-CME solver (with $10^4$ samples) and the Monte Carlo method (with $10^5$ samples) in estimating the marginal distributions. Both approaches have relatively the same computational time, and they both accurately estimate the marginal distributions of individual species. The superior performance of the RB-CME solver is attributed to the estimation of the follower system, which dominates the whole estimation problem. Source data are provided as a Source Data file.

to the fact that both the RB-CME solver and the Monte Carlo method use the same protocol to estimate the leader system, and for the same time cost, the RB-CME solver has fewer samples than the Monte Carlo method. Despite this issue, the RB-CME solver still has a much better performance in estimating the whole system because of the greater benefit gained from the estimation of the follower part.

## Application of the RB-CME solver to the repressilator

To demonstrate the performance of the RB-CME solver for nonlinear systems, we consider a well-known genetic circuit called the repressilator (see Fig. 5), where three gene expression systems cyclically repress each other. The repressilator was first engineered by Elowitz and Leibler in 2000[18], and its name comes from the cyclical repression topology and its oscillatory dynamic behavior. In this system, the nonlinearity comes from the mRNA production processes, whose propensities are Hill functions. More details about the modeling are presented in Supplementary Information, Section S6.A.

In this model, we intend to compare the RB-CME solver with other approaches. Here, our primary objective is to estimate the joint distribution of the chemical species, as this offers crucial insights into the interacting behaviors of the species driven by cyclic repression. With the parameters set in Supplementary Information, Section S6.A, most of the probability is contained in the state space where each mRNA has no more than 20 copies, and each protein has no more than 200

copies. Notice that storing a probability distribution on this state space requires about 500 Gigabytes of memory (8 bytes/state × $\left(20^3 \times 200^3\right)$ states), so applying the FSP approach to this example is impractical. Alternatively, to get an accurate solution of the CME, we simulated $3 \times 10^9$ trajectories of this dynamical system and used the empirical distribution to approximate the exact probability distribution. This whole procedure took 10 graphics processing units (GPUs) about 24 hours. For the RB-CME solver, we set the truncated state space for each mRNA to be $\{0, 1, \ldots, 19\}$ (if the mRNA is classified as a follower-level species), the truncated state space for each protein to be $\{0, 1, \ldots, 199\}$ (if the protein is classified as a follower-level species), and each follower subsystem to contain no more than 200 states. In this setting, the leader-follower decomposition algorithm classifies all the mRNAs as the leader-level species and all the proteins as the follower-level species. Finally, we applied both the RB-CME solver (with the filtered FSP as the chosed filtering approach) and Monte Carlo method to the repressilator; the results are shown in Fig. 5.

Again, the results show that the RB-CME solver is efficient and accurate in this example. Specifically, Fig. 5B tells that given the same sample size, the RB-CME solver is about 15 times more accurate than the Monte Carlo method, and given the same time cost, the RB-CME solver is 4 times more accurate. Notably, by only taking one CPU and about half an hour, our method can provide a very accurate solution with an $L_1$ error of 3% in estimating marginal distributions of the mRNAs and the proteins. Furthermore, we compared the performance of both approaches in estimating the marginal distributions when their computational time is relatively the same (see Fig. 5C). From this panel, we can observe that both methods accurately estimate the marginal

distribution of individual species, and the superior performance of the RB-CME solver is attributed to the estimation of the follower system. More specifically, the proteins have much more molecular counts than the mRNAs, and, therefore, the marginal probability distribution of the proteins tends to be more dispersed than that of the mRNAs. According to (4) and its relevant analysis, this explains why the estimation error of the follower system (the proteins) dominates in the Monte Carlo method (see Fig. 5C). In contrast, the RB-CME solver has a much lower error in estimating the proteins because it applies a FSP method to the follower system, and this advantage greatly dominates the disadvantage of the RB-CME solver in estimating the mRNAs (see Fig. 5C). More interestingly, the improvement of the RB-CME solver in estimating the whole follower system is much greater than that in estimating each follower subsystem (see Fig. 5C) thanks to our modularization strategy, which further demonstrates the scalability of our approach with increasing system dimensions. These observations also imply that the RB-CME solver can accurately estimate other genetic circuits if all the species with large molecular counts are classified as follower-level species.

## Applying the RB-PF to stochastic filtering for the genetic toggle switch

Now, we consider the filtering problem for another well-known genetic circuit, called the genetic toggle switch, which was first engineered by Gardner, Cantor, and Collins in 2000[20]. This circuit consists of two gene expression systems, whose protein products repress each other's expression (see Fig. 6A), and their trajectories exhibit switching behaviors (see Fig. 6B). We assume that the first protein is fluorescent and measured by a microscope at several time points, and our goal is to infer the hidden dynamical states given the observations. More details about the modeling are presented in Supplementary Information, Section S6.B.

In this example, we intended to compare the RB-PF with other filtering approaches. First, we simulated a trajectory of the system and generated observations at 10 different time points. Our task was to infer the hidden states based on these generated observations. To get an accurate approximation of the exact filter, we applied the FSP to the problem with a state space where each protein has fewer than 200 copies. This procedure took 6.5 h on a 12-core CPU. For the RB-PF, we set the truncated state space for each protein to be $\{0, 1, ..., 199\}$ (if it is classified as a follower-level species) and the truncated state space for genes to be $\{0, 1\}$ (if it is classified as a follower-level species). By letting each follower subsystem have no more than 200 states, our approach classifies all the proteins as the follower-level species and the rest as leader-level species. Finally, we applied both the RB-PF and conventional particle filter to the filtering problem. In the prediction step, this RB-PF uses an RB-CME solver that adopts the filtered FSP as the selected filtering algorithm in its framework. The results are shown in Fig. 6.

In Fig. 6, we compare the performance of the RB-PF ($10^4$ samples) and PF ($10^5$ samples), which have the similar time cost (see Fig. 6G). The numerical results show that though the RB-PF and the PF have similar performance in estimating the conditional mean and variance of the second protein (Fig. 6B), the RB-PF is far more accurate in estimating the whole conditional probability distribution. Specifically, the RB-PF consistently performs better than the PF at different observation time points, and the improvement ratio is always significant (Fig. 6C). When looking at the marginal conditional distributions, we can observe that the superior performance of the RB-PF also comes from the estimation of the follower subsystems (Fig. 6D). Particularly, in this example, the estimation error of the follower system (the proteins) dominates in the PF (see Fig. 6D), as its conditional probability distribution is more dispersed than that of the leader system. Thanks to the modularization strategy and the filtered FSP that our method applies to the follower system, the RB-PF is more accurate than the PF in estimating the

follower system (see Fig. 6D), and, therefore, the RB-PF is significantly more accurate in estimating the whole conditional probability distribution. From Fig. 6E, we can observe that for the follower system, the solution of the RB-PF is more smooth and accurate than the solution of the PF, demonstrating the advantage of the RB-PF. More interestingly, in this filtering problem, the RB-PF and the PF have similar performance in estimating the leader system (Fig. 6D, E), quite different from the situation in solving the CME where the Rao-Blackwell method is less accurate than the conventional Monte Carlo method in estimating the leader (see Figs. 4 and 5 in previous examples). The reason for this is that in the adjustment step of the filtering algorithm, the conditional probability distribution of the leader system is influenced by the conditional probability distribution of the follower system via the likelihood function, and, therefore, the leader system can also benefit from Rao-Blackwellization.

We also compared the performance of the RB-PF and the PF under various different conditions. First, the performance of both filters is quite robust to the variation of the observation noise intensity Fig. 6F. Moreover, the RB-PF and the RB-CME solver perform similarly in their associated problems (see Fig. 6G and H); the same is true for the PF and the Monte Carlo method. These results agree well with our theoretical analysis (also see Supplementary Information, Section S2.B and Section S4.C). Also, they suggest the consistency result that if the RB-CME solver can accurately solve a CME, then the RB-PF should also perform well in the associated filtering problem. In addition, similar to the situation in solving the CME, the RB-PF is orders of magnitude more accurate than the PF given the same sample size or the same time cost (see Fig. 6G). All these conclusions indicate the reliability of our approach.

## Identifying transcription dynamics in Yeast cells from single-cell time-course trajectories

In the following, we applied our identification algorithm to an experimental dataset of yeast cells and used the result to analyze the sources of cell-to-cell variability.

Cell-to-cell heterogeneity is influenced by two main factors: the random firing of reactions within individual cells (known as intrinsic noise) and the variability in the parameters that determine the reaction propensities across the population (known as extrinsic noise). Understanding the role of intrinsic and extrinsic noise in shaping the cell-to-cell variability is an important topic in biology. We investigated this issue in the context of transcription dynamics in yeast cells, using our parameter identification method introduced in this paper.

We focused on genetically identical cells engineered in[90] (see Fig. 7A for an illustration of the synthetic circuit). Within this gene circuit, VP-EL222 homodimerizes in the presence of light, which then binds to its cognate promoter (a fusion of several EL222-binding sites) to stimulate the expression of a downstream gene. The produced RNAs contain stem-loops that can be recognized and bound by a fluorescent reporter (tdPCP-tdmRuby3); this structure allows for the measurement of RNA dynamics under a microscope.

A reaction network model for this cell system is presented in Fig. 7B. Since the DNA contains several sites, a simple telegraph model containing only two gene states (ON and OFF) may not adequately represent this practical system. Consequently, we considered a three-gene-state model, where the gene has one inactive state and two active states. In these different active states, the RNAs are transcribed at different rates. Particularly, when the reaction rate $k_3$ equals zero, this model reduces to the conventional telegraph model. This mRNA transcription dynamics also invalidates the applicability of the method in[11], because the method requires the systems to have distinct reaction vectors when expressed by mass-action kinetics, which is not the case for the mRNA transcription dynamics.

Here, we aimed to identify the dynamical parameters of each cell from the time-trajectory of its RNA dynamics and then investigate the

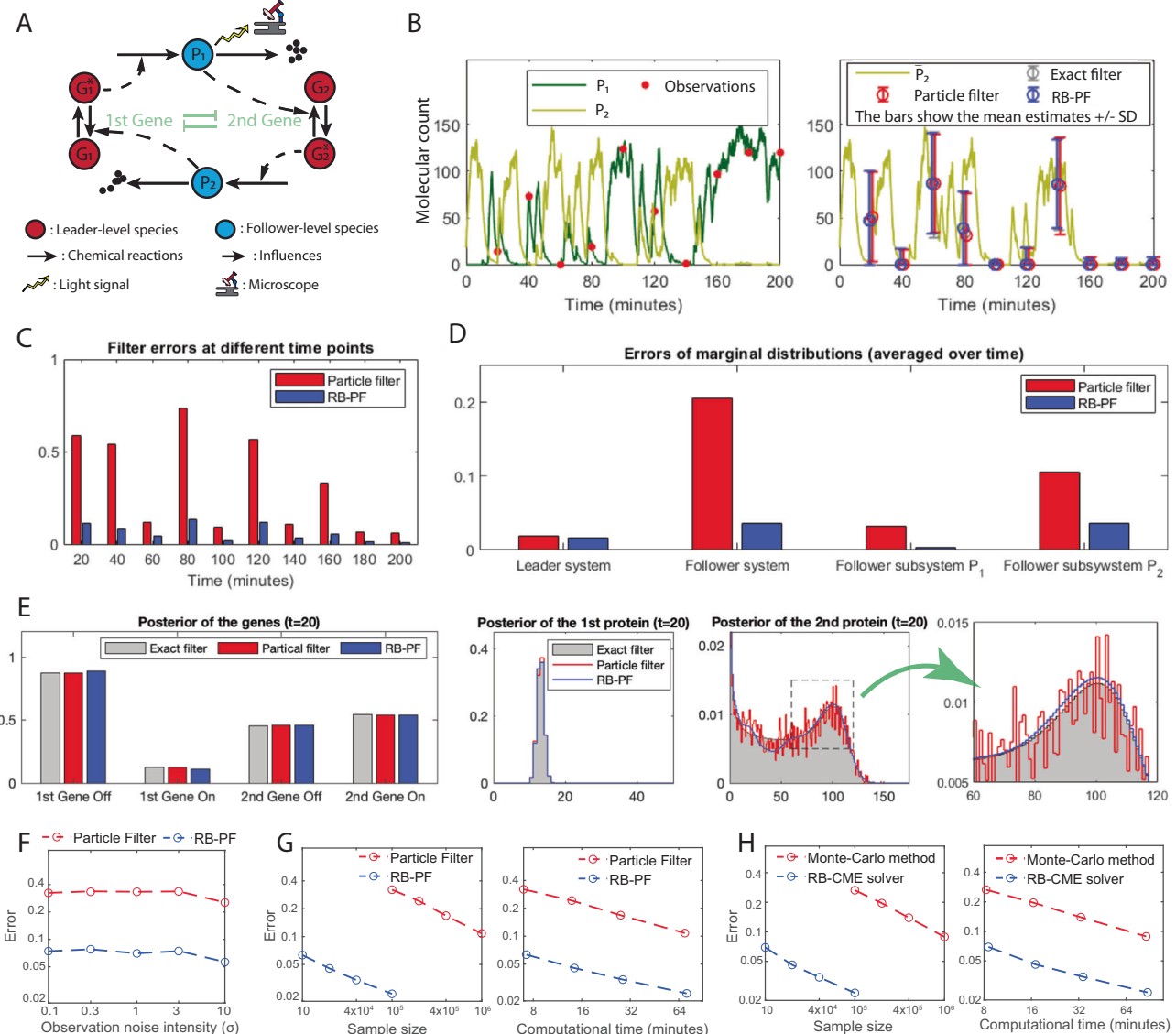

**Fig. 6 | Filtering results for the genetic toggle switch. A** Genetic toggle switch where two gene expression systems repress each other. The first protein is fluorescent, and its trajectory is measured with the observation noise intensity of 1. Our approach classifies all the genes as the leader-level species and all the proteins as the follower-level species. **B** Performance of different filters in estimating the conditional mean of the second protein given a discrete-time trajectory of the first protein. The error bars represent the interval of the conditional mean ± conditional standard deviation (SD). The exact filter is approximated by the FSP method. The particle filter (PF) and the Rao-Blackwellized particle filter (RB-PF) have $10^5$ samples and $10^4$ samples, ensuring that their computational time is similar. This panel shows that all the filters have similar performance in estimating the conditional mean of the second protein. **C** $L_1$ errors of the PF and the RB-PF at different time points. **D** Average $L_1$ errors of the filters in estimating the marginal distributions. **E** Evaluation of marginal conditional distribution at the first observation time

($t = 20$). Panels (**C**) to (**E**) show that the RB-PF significantly outperforms the PF in terms of the $L_1$ error, and the major advantage lies in the estimates of the follower system. **F** Performance of the filters under different observation noise intensities. The PF and RB-PF have, respectively, $10^5$ samples and $10^4$ samples, ensuring that their computational time is similar. The performance is evaluated by the average $L_1$ error over 10 observation time points; the same applies in panels (**G**) and (**H**). This panel shows that the performance of both filters is robust to the intensity of the observation noise. **G** Convergence of the filters under a fixed observation noise intensity ($\sigma = 1$). **H** Convergence of the Monte Carlo method and the RB-CME solver in computing the CME of the genetic toggle switch at time 200. Panels (**G**) and (**H**) show that the RB-CME solver and RB-PF have similar performance in their associated problems given the same sample size or the same computational time. The same result also applies to the Monte Carlo method and PF. Source data are provided as a Source Data file.

contribution of intrinsic and extrinsic noise to the overall cell-to-cell variability. In particular, we compare the intrinsic and extrinsic noise estimates obtained from our cell-specific inference results with the estimates obtained directly from the time-course experimental data. To perform this direct estimation, we propose a novel decomposition technique which does not require dual-reporters[77,79,80] but produces equivalent results under the assumption of stability of the underlying stochastic reaction network (see the section *"Analysis of noise decomposition for the yeast cells"*).

The code for the analysis in this transcription system is available on GitHub: "https://github.com/ZhouFang92/Rao-Blackwell-method-for-cell-specific-model-identification".

### In-silico verification of the proposed identification method

First, we examined the accuracy of our Rao-Blackwell method in identifying model parameters through numerical simulation. In this simulation study, we assumed that $k_1, ..., k_4$ can take values from the set $\{0, 0.05, ..., 1\}$, and $k_{p_1}, k_{p_2}$ were within the set $\{1, 2, ..., 10\}$. All these

A

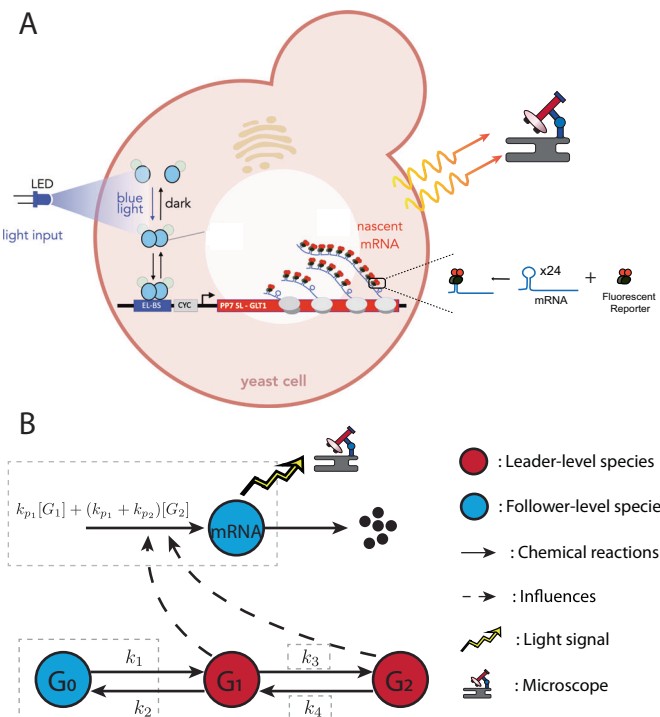

B

**Fig. 7 | Transcription system in Yeast cells. A** Synthetic circuit in yeast cells. In the presence of light, EL222 can dimerize, bind to EL222-binding sites (EL-BS), and activate the expression of the downstream gene. The transcribed RNAs contain stem-loops to which fluorescent reporters can attach, allowing visualization of RNA dynamics. **B** Reaction network model for the gene circuit. In this model, the gene has three states: one inactive state $G_0$ and two active states $G_1$ and $G_2$. In the first active state $G_1$, the mRNA is transcribed at a rate $k_{p_1}$. In the second active state $G_2$, the mRNA is transcribed at a higher rate $k_{p_1} + k_{p_2}$, where $k_{p_2}$ represents an additional rate of transcription beyond $k_{p_1}$. The parameters $k_1, ..., k_4, k_{p_1}$, and $k_{p_2}$ are unknown but fixed over time, and the mRNA degrades at a linear rate with the rate constant 1. Under the setting in this paper, our algorithm consistently classifies $G_1$ and $G_2$ as leader species, with the remaining elements categorized as follower components. The follower part has four subsystems: the first subsystem consists of $G_0$, $k_1$, and $k_2$, the second subsystem consists of $k_3$, the third subsystem consists of $k_4$, and the last subsystem consists of $k_{p_1}$, $k_{p_2}$ and mRNA.

parameters were in units of minute$^{-1}$ and had uniform prior distributions, with the exception of $k_3$ which represents the rate of switching from the first active state $G_1$ to the second $G_2$. For $k_3$, half of its initial probability mass was allocated at zero to indicate the uncertainty of whether the number of active gene states is one or two, and the rest of the probability was uniformly distributed over states 0.05, 0.1, ..., 1. At the initial time, we set the gene state to $G_0$ and the mRNA count to zero. Moreover, the mRNA count was measured every minute with an observation noise intensity of 0.1.

We first generated a simulated trajectory of the given model with randomly selected parameters. Then, we applied our Rao-Blackwell method to inferring these model parameters using the simulated time-course measurements of this system. We truncated the space for the mRNA count to be {0, 1, ..., 20} and that for each gene to be {0, 1}; also, we set the sample size in our algorithm to 10,000. By requiring the size of the maximum follower subsystem to be less than 30,000, our algorithm classified $G_1$ and $G_2$ as leader species and assigned the remaining components into four follower subsystems (see Fig. 7B for the decomposition). Moreover, in prediction steps (10), our method uses an RB-CME solver that adopts the filtered FSP as the selected filtering algorithm in its framework. The numerical result of our algorithm is presented in Fig. 8A.

The gray box in Fig. 8A illustrates that in this inference problem, our algorithm accurately identifies the model parameters, with the maximum a posteriori (MAP) estimates being exactly or very close to the real values. Moreover, the provided conditional distributions of the parameters are quite narrow, indicating that these estimates are relatively confident. We have also tested and observed the good performance of our method with different sets of actual parameter values (see Supplementary Information, Section S7.C and Fig. S1). In conclusion, our inference method is quite effective in this simulation example.

In practical applications, the actual parameter values for a specific cell are usually unknown, which makes it impractical to validate the maximum a posteriori (MAP) estimate by directly comparing it with the true values. Consequently, there is a need for a method to validate the identification results using experimental data, without the necessity for precise parameter information. To this end, we propose a verification method that compares the stationary probability distributions of the target cell and the inferred model. Note that the stationary probability distribution of the target cell is still not directly available; by definition, it requires the measurement of a cell population having identical parameters to the target cell. Fortunately, the considered system is stable in the ergodic sense[91] (see Supplementary Information, Section S7.B), and consequently the occupation time distribution $P_{oc}(T, x) \triangleq \frac{1}{T} \int_0^T \mathbb{1}(X(t) = x) dt$ converges to the stationary distribution as $T \to \infty$. Thus, the stationary distribution of a target cell system can be approximated by its occupation time distribution, which is available from the mRNA measurements. Moreover, to account for the measurement noise, we rounded each measurement to the nearest integer and used the rounded numbers to compute the occupation time distribution. Meanwhile, the stationary distribution of the inferred model is approximated by a distribution computed by the FSP at a large time point. The time point is chosen so that the stationary distribution is approximately reached. In conclusion, this verification method, through comparing the occupation time distribution (of the target cell) and the stationary distribution (of the inferred model), is feasible in the experimental setting.

The comparison of these two distributions is presented in the bottom-left panel of Fig. 8A. We can observe that the two distributions almost overlap perfectly, with a Kullback-Leibler divergence of 0.02. The slight difference might be attributed to the small difference between the MAP estimates and the true parameter values (see the gray box in Fig. 8A) and the imperfect approximation of the stationary distribution via the occupation time distribution. In general, the closely matched distributions suggest that our identification approach is accurate in this example.

Next, we tested whether our approach could correctly identify the model when the target system had only two gene states, i.e., $k_3 = 0$. Similar to the previous case, we first simulated a trajectory of the system with parameters $k_1, k_2$, and $k_{p_1}$ identical to the values selected previously ($k_1 = 0.3, k_2 = 0.4, k_{p_1} = 3$), while setting $k_3 = k_4 = k_{p_2} = 0$. Then we used our Rao-Blackwell method to identify the model via the simulated measurements. We maintained the same settings for the identification algorithm as in the previous numerical example in this section. The result, presented in Fig. 8B, shows that our algorithm accurately infers the parameters $k_1, k_2, k_3$, and $k_{p_1}$, with the MAP estimates either matching or closely resembling the true value. Notably, the algorithm provided a high-confidence estimate of $k_3 = 0$ with a conditional probability of 0.9 and therefore successfully recognized the correct two-gene-state model of the system. Since $k_4$ and $k_{p_2}$ have no effect on the dynamics of the two-gene-state model, the inference of these parameters is unimportant in this context, and quite expectedly, our algorithm yields conditional distributions close to uniform distributions (the prior distribution). Moreover, the stationary distributions of the target system and the inferred model also match closely (see the bottom-left panel in Fig. 8B), suggesting the accuracy of the inferred model. To conclude, our approach can accurately identify the dynamical model when the target system has only two gene states.

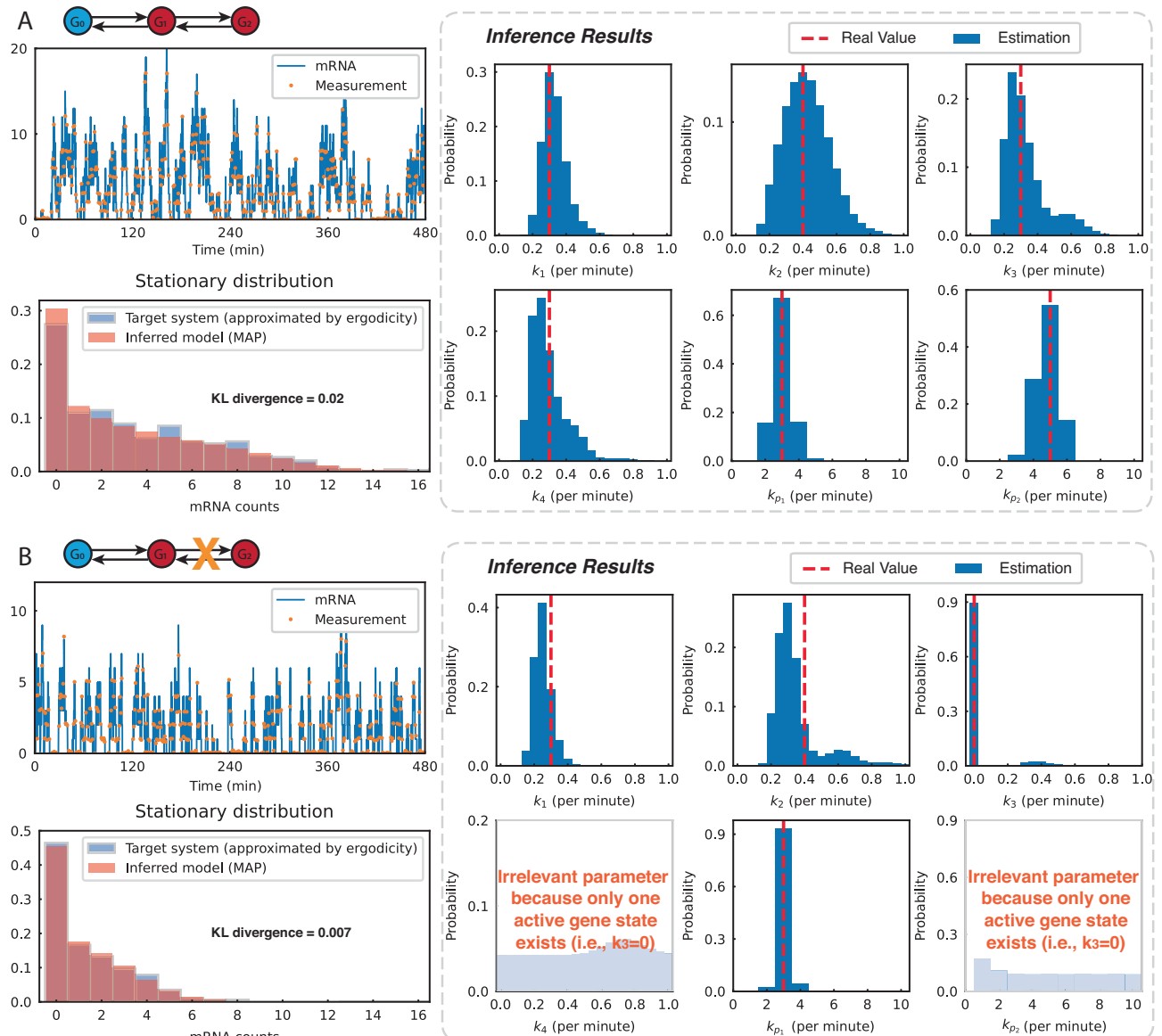

**Fig. 8 | Performance of the Rao-Blackwell identification algorithm on the yeast transcription system with simulated data. A** Inference of a 3-gene-state system. We first simulated a 3-gene-state system (see Fig. 7B) with parameters $k_1 = 0.3$, $k_2 = 0.4$, $k_3 = 0.3$, $k_4 = 0.3$, $k_{p_1} = 3$, and $k_{p_2} = 5$ (all in units per minute); its mRNA dynamics and time-course measurements are depicted in the top-left panel. Next, we utilized our algorithm to identify the model parameters using the simulated measurements. The sample size of our algorithm was set to 10,000, and the result is presented in the box surrounded by the dash lines. The results illustrate that our algorithm accurately infers the hidden model parameters. The bottom-left panel compares the stationary distributions of the target system and the inferred model (with parameters being the maximum a posteriori estimates). Due to ergodicity, the stationary distribution of the target system was approximated by the occupation time distribution of the mRNA measurements. This bottom-left panel shows a close match between the two stationary distributions, suggesting the accuracy of the

inferred model. **B** Inference of a 2-gene-state system. We tested our approach when the system had only two gene states. We first simulated the system with parameters the same as before, except that $k_3 = k_4 = k_{p_2} = 0$, i.e., the system had only one active gene state. The mRNA dynamics and its time-course measurements are presented in the top-left panel. Then, we used our algorithm to identify the model parameters with the sample size set to 10,000; the result is presented in the box surrounded by the dash lines. The results illustrate that our algorithm accurately infers the parameters $k_1$, $k_2$, $k_3$, and $k_{p_1}$; as the maximum a posteriori estimate of $k_3$ is zero, our algorithm correctly identifies the two-gene-state model. For the irrelevant parameters $k_4$ and $k_{p_2}$, our algorithm expectedly gives conditional distributions close to uniform distributions (the prior distribution). The bottom-left panel shows a close match between the stationary distributions of the target system and the inferred model, suggesting the accuracy of the identification result. Source data are provided as a Source Data file.

Our algorithm is also applicable to the inference problem when the observation noise intensity $\sigma$ is unknown. In this case, this $\sigma$ can be viewed as an unknown parameter and inferred simultaneously with other model parameters ($k_1, k_2, k_3, k_4, k_{p_1}$, and $k_{p_2}$). We performed numerical experiments to test the performance of our algorithm in this situation (see Supplementary Information, Section S7.D and Fig. S2). The results indicate that our method can accurately estimate the observation noise

intensity $\sigma$ as well as other parameters with the presence of $\sigma$ uncertainty.

**Parameter identification for yeast cells from experimental data**
We investigated the performance of our identification method in experimental data and further applied this to understanding the contribution of intrinsic and extrinsic noise to the cell-to-cell dynamical variability (shown in the next section).

The experiment used the yeast cell constructed in ref. 90. As mentioned in ref. 90, all strains were derived from BY4741 and BY4742 (Euroscarf, Germany). Before the experiments, all the cells were kept in a dark environment, ensuring the gene started in the inactive state and the mRNA count to be initially zero. Then, the cells were placed under a microscope platform developed in[90,92], and they were stimulated by being exposed to constant light. Subsequently, their mRNA fluorescence was measured every 2 min for a total period of 4 hours. Finally, this experimental process resulted in the collection of single-cell time-course data from 130 cells. Due to background noise, the platform can only provide the readout when the mRNA count is greater than 7, i.e., $h(x) = x\mathbb{1}(x > 7)$ (see (7) for the meaning of $h(\cdot)$). Moreover, the mRNA measurements provided by this platform are considered fairly accurate, though the specific magnitude of the measurement noise has not been exactly quantified in the literature[90,92]. Therefore, we treated the measurement noise intensity $\sigma$ as 1.

To account for our limited knowledge about the parameter values, we considered large ranges for the values of the parameters. Specifically, we assumed $k_1$, $k_2$, and $k_3$ to be within the set $\{0, 0.05, ..., 1\}$, $k_4$ within the set $\{0, 0.1, ..., 2\}$, $k_{p_1}$ within $\{0, 8, 16, ..., 80\}$, and $k_{p_2}$ within $\{20, 30, 40, ..., 120\}$. All these parameters were assumed to have uniform prior distributions over their specified range, with the exception of $k_3$. For $k_3$, we assumed that half of its prior probability mass is concentrated at state zero, reflecting the uncertainty regarding the actual number of gene states; the rest of the mass was uniformly distributed on the remaining values in its range. Also, we truncated the state space for the mRNA count to be $\{0, 1, ..., 150\}$ and that for each gene ($G_0$, $G_1$, and $G_2$) to be $\{0, 1\}$. Moreover, the sample size of our algorithm was set to 3000, and the size of each follower subsystem was required to be less than 30,000. Under this setting, our algorithm classifies $G_1$ and $G_2$ as leader species and assigns the remaining components into four follower subsystems (see Fig. 7B for the decomposition). Some of the identification results are presented in Fig. 9.

Figure 9A presents the inference results for cell #78, randomly selected among the cells exhibiting sufficiently active transcription processes. Our algorithm provides confident estimates for each parameter of the cell system, as evidenced by the relatively narrow conditional distributions. To verify the estimation result, we also compared the stationary distributions of the inferred model and the actual cell system, using the approach introduced in the preceding section. The result shows that these two distributions both have a bimodal shape and align well with a KL divergence of 0.031 (see the top-middle panel of Fig. 9A). We noticed that the stationary distribution is sensitive to the parameters, with the elasticity (defined by $\frac{\partial \mathbb{E}_{st}[X_{mRNA}|\theta^{inf}] / \mathbb{E}_{st}[X_{mRNA}|\theta^{inf}]}{\partial \theta_i^{inf} / \theta_i^{inf}}$ for each inferred parameter $\theta_i^{inf}$) being 35%,-36%, 32%,-30%, 69%, and 31% for $k_1$, $k_2$, $k_3$, $k_4$, $k_{p_1}$, and $k_{p_2}$, respectively. Therefore, this consistency between the stationary distributions indicates that the inferred parameters are consistent with the true system parameters. The slight difference between these distributions can be attributed to the imperfect approximation of the stationary distribution (of the actual system) by the occupation time distribution and the imperfect inference result due to the limited length of the mRNA trajectory. It can also be attributed to the coarse discretization scheme in the parameter space. We have also proposed an improved RB-PF based on grid refinement, which can reduce this discrepancy by one-third (Supplementary Information, Section S7.E and Fig. S3). Moreover, by applying this improved RB-PF, we also confidently estimated the observation noise to be 0.4, suggesting that the mRNA measurement is even more accurate than what we previously assumed (i.e., $\sigma = 1$) (see Supplementary Information, Section S7.F and Fig. S4). However, this more accurate estimation of $\sigma$ does not significantly impact the inference results for the other parameters and, therefore, does not contribute to further reduction of the discrepancy between the

inferred model and the actual system. Overall, all the results illustrate the accuracy of our inference results.

The identification result indicates that the parameter $k_3$ was positive with a high probability and, therefore, suggests that the system had three gene states. To further validate this result, we also inferred the system, assuming only two gene states existed. In this identification problem, we set $k_3 = k_{p_2} = 0$, considered $k_{p_1}$ within the set $\{0, 1, ..., 121\}$, and kept the other settings as before. We also compared the stationary distributions of the inferred model and the actual cell system and found that the two distributions had a big mismatch (see the gray box in Fig. 9A). Specifically, in this case, the inferred model fails to capture the bimodal distribution of the actual cell system, and the KL divergence between these two distributions is relatively large, with a value of 0.145 (about five times the value in the previous case). All these results support the validity of the three-gene-state model for the real system.

Figure 9 B shows the inference results for some typical cells in the population. These cells displayed a variety of behaviors: some cells spent the majority (≥90%) of the time in the inactive gene state, while others spent only half of their time in this state. Some cells were highly active, exhibiting a unimodal stationary distribution with a peak away from the origin, while others showed switch-like dynamics, exhibiting a bimodal stationary distribution. Despite this variability, our algorithm consistently provides accurate inference results for all the cells, as evident by the close match between the stationary distributions of the inferred models and the actual cells. These results demonstrate the effectiveness of our method in identifying dynamical models of real biological cells from their individually measured time-trajectories.

## Analysis of noise decomposition for the yeast cells

The inference results show that the cells displayed significant heterogeneity in their system parameters (see Fig. 10A). We then employed the inference result to investigate how this heterogeneity affected the variability of mRNA counts in the cell population and whether this is a dominant noise source.

The total variability of mRNA counts (at the stationary probability distribution) across the cell population is $\text{Var}\left(X_{mRNA}^*\right)$. Here, Var is the notation of variance, $X_{mRNA}^*$ represents the mRNA count (at the stationary probability distribution) of a randomly selected cell, and $\theta$ is a vector of system parameters of this selected cell. As discussed in[77,79] we can apply the law of total variance to decompose this variability or *noise* into extrinsic and intrinsic components as

$$\underbrace{\text{Var}\left(X_{mRNA}^*\right)}_{\text{total noise}} = \underbrace{\mathbb{E}\left[\text{Var}\left(\mathbb{X}_{mRNA}^*|\theta\right)\right]}_{\text{intrinsic noise}} + \underbrace{\text{Var}\left(\mathbb{E}\left[X_{mRNA}^*|\theta\right]\right)}_{\text{extrinsic noise}}.$$

The first term on the right is the intrinsic noise, because in the conditional variance $\text{Var}(X_{mRNA}^*|\theta)$, the parameter $\theta$ is fixed and so only the noise due to the random firing of reactions contributes to this term. On the other hand, the second term quantifies the extrinsic noise because the conditional expectation $\mathbb{E}\left[X_{mRNA}^*|\theta\right]$ filters out the noise from the random firing of reactions, and hence its variance quantifies the noise-contribution due to the variability in the extrinsic parameter $\theta$.

Direct estimation of these two terms from experimental single-cell data is difficult as computing the conditional expectation and variance, given a fixed $\theta$-value, would require us to have measurements from a population of cells that share this $\theta$-value. As we cannot have such measurements, the strategy proposed in the current literature to measure the two noise components relies on having dual-reporters within each cell[77,79,80]. These two reporters should not only provide conditionally independent measurements given $\theta$, but also these measurements must have the same first two moments so that their covariance becomes an estimate of the extrinsic noise. These two conditionals make

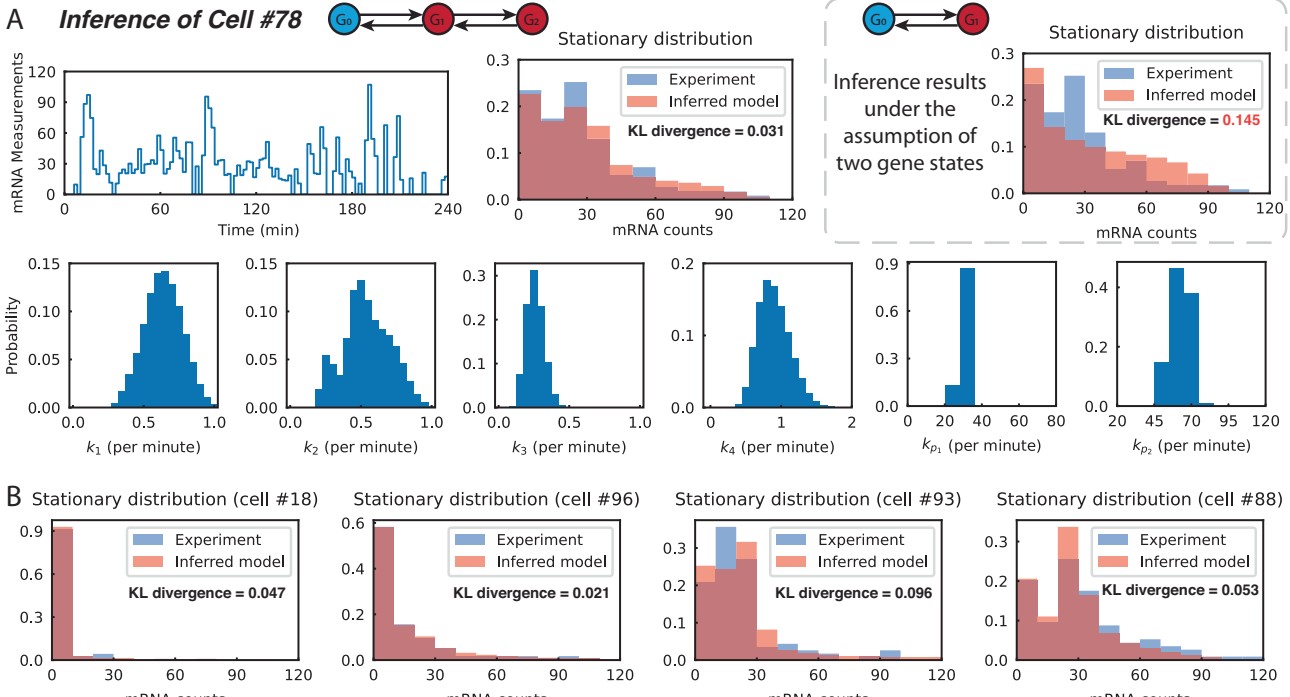

**Fig. 9 | Performance of the Rao-Blackwell identification algorithm in experimental data from yeast cells. A** Inference result of cell #78. The top-left panel shows mRNA dynamics measured every two minutes over a 4-hour period. The lower row presents the parameter estimates, with the narrow conditional probability distributions illustrating the high confidence of these estimates. The top-middle panel compares the stationary distributions of the inferred model (using the MAP estimates) and the actual cell (approximated by the occupation time distribution of mRNA measurements). In these distributions, states have been organized in groups of ten. The good agreement between these two distributions (evident from the bimodal structure and low KL divergence) underscores the accuracy of the identification result. Our results indicate that $k_3$ is positive, implying that the system comprises three gene states. We also inferred the system assuming only two gene states; these results are shown in the gray box. In this case, the

stationary distribution of the inferred model does not align with the distribution of the real cell. Specifically, the inferred model fails to capture the bimodality of the actual stationary distribution, leading to a relatively substantial KL divergence of 0.145 between the two distributions. This observation further supports the validity of the three-gene-state model for the real cell. **B** Inference results of some typical cells. We present the inference results of several cells exhibiting different behaviors. Cell #18 took about 90% of the time staying in the inactive gene state; cell #96 had a shorter duration in the inactive gene state; cell #93 took even less time in the inactive state, and its stationary distribution has a peak different from the origin; cell #88 exhibited a bimodal stationary distribution. In all these cases, the stationary distribution of the inferred model closely matches that measured in the experiment in terms of the shape and KL divergence, indicating the validity of our algorithm. Source data are provided as a Source Data file.

dual-reporter systems hard to realize, and therefore it is of interest to find other ways to directly estimate noise components from single-cell experimental data without needing such dual-reporter systems.

We now propose a approach to measure the two noise components, in the situation where the underlying stochastic dynamics is stable (i.e. ergodic), and we have time-course experimental data for each measured cell. In this case the required conditional expectation and variance (given $\theta$) can be approximated by time-averages over a large interval $[0, T]$, i.e. $\mathbb{E}\left[X^*_{\text{mRNA}}|\theta\right] \approx \frac{1}{T}\int_0^T X^\theta_{\text{mRNA}}(t)\mathrm{d}t$ and $\text{Var}\left(X^*_{\text{mRNA}}|\theta\right) \approx \frac{1}{T}\int_0^t \left(X^\theta_{\text{mRNA}}(t)\right)^2\mathrm{d}t - \left(\mathbb{E}\left[X^*_{\text{mRNA}}|\theta\right]\right)^2$, where $X^\theta_{\text{mRNA}}(t)$ is the mRNA dynamics of a cell with parameters $\theta$ (see Supplementary Information, Section S8 for more details). Therefore extrinsic and intrinsic noise can be estimated as

$$\text{Extrinsic noise} \approx \text{Var}\left(\frac{1}{T}\int_0^T X^\theta_{\text{mRNA}}(t)\mathrm{d}t\right) \qquad (13)$$

$$\text{Intrinsic noise} \approx \mathbb{E}\left[\frac{1}{T}\int_0^T \left(X^\theta_{\text{mRNA}}(t)\right)^2\mathrm{d}t - \left(\frac{1}{T}\int_0^T X^\theta_{\text{mRNA}}(t)\mathrm{d}t\right)^2\right]. \qquad (14)$$

We employed this method to evaluate the intrinsic and extrinsic noise in the yeast cell population, and the result is presented in Fig. 10.B and C (labeled as "experiments"). Our cell-specific

identification results provide an indirect inference-based approach for the estimation of the two noise components. Specifically, our inference result provided cell-specific estimates for the parameters, thereby allowing for the computation of $\mathbb{E}\left[X^*_{\text{mRNA}}|\theta\right]$ and $\text{Var}\left(X^*_{\text{mRNA}}|\theta\right)$ for each cell. Then, by combining these conditional mean and variance, we obtained the estimates of intrinsic and extrinsic noise, as illustrated in Fig. 10C (labeled as "inferred models").

The noise decomposition results obtained from both methods are quite consistent (see Fig. 10), indicating the accuracy of our identification method and the reliability of the noise decomposition result. The result in Fig. 10C suggests that the extrinsic noise only accounted for a small fraction (about 18%) of the total variation, though the heterogeneity in dynamic parameters was significant. This result is mainly attributed to the fact that the intrinsic noise in this transcription dynamics was very large. Figure 9A tells that the mRNA dynamics of the 78th cell fluctuated greatly from 0 to 100, with the coefficient of variation (defined as $\frac{\text{standard deviation}}{\text{mean}}$) being almost one. This fluctuation of mRNA dynamics was also observed in the other cells, as a result of which the intrinsic noise contributes to 82% of the total noise (Fig. 10C). In contrast, the extrinsic noise only contributes to about 18% of the total noise (Fig. 10C). Overall, in this transcription dynamics, the intrinsic noise dominated the total variation.

## Discussion

The modeling and analysis of intracellular reactions, which are subject to inherent randomness in living cells, is typically achieved through

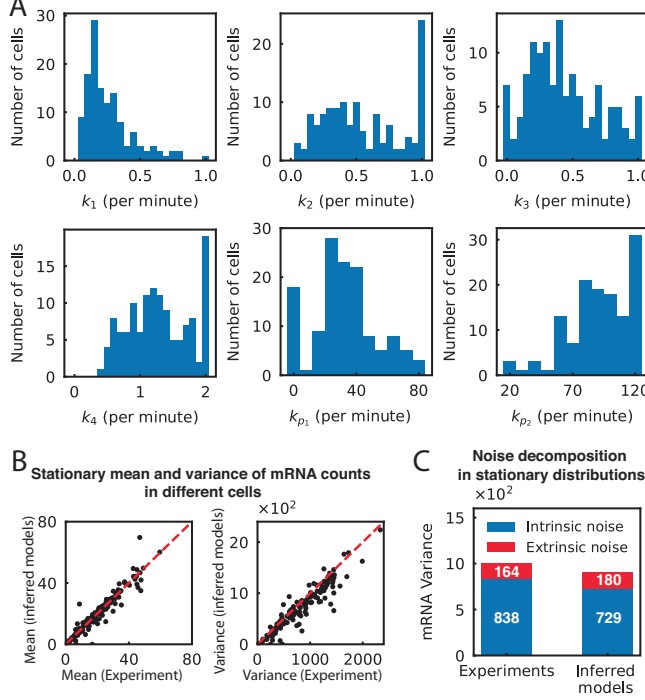

**Fig. 10 | Analysis of noise in gene expression of Yeast cells. A** Distribution of inferred parameters across the cell population. The plots indicate significant variability in system parameters within the yeast population. **B** Stationary mean and variance of mRNA counts in different cells. We contrast the mean and variance estimates from the inferred models with those estimated from experimental data. The stationary distribution of each real cell was estimated using the occupation time distribution of mRNA measurements. The results show that our estimates are consistent with the experimental data, with all the dots distributed around the diagonal lines. Also, the results demonstrate that the yeast cells displayed considerable heterogeneity in stationary mean and variance. (C) Noise decomposition in stationary distributions. The variance of mRNA counts across the population can be expressed as $\mathrm{Var}(X_{\mathrm{mRNA}}) = \mathbb{E}[\mathrm{Var}(X_{\mathrm{mRNA}}|\theta)] + \mathrm{Var}(\mathbb{E}[X_{\mathrm{mRNA}}|\theta])$, where $\mathbb{E}$ and Var are the notations of mean and variance under the stationary distribution, $X_{\mathrm{mRNA}}$ is the mRNA count of a randomly selected cell, and $\theta$ represents the model parameters of that cell. $\mathbb{E}[\mathrm{Var}(X_{\mathrm{mRNA}}|\theta)]$ and $\mathrm{Var}(\mathbb{E}[X_{\mathrm{mRNA}}|\theta])$ are the intrinsic noise and extrinsic noise. We compare these types of noise obtained from the experimental data and inferred models. The plot shows consistency between the noise decomposition results derived in both ways with relative discrepancies about 10%. These discrepancies can be attributed to the limited duration of the measurement period (bringing in bias in the estimates), the exclusion of certain biological mechanisms (e.g., cell cycles), etc. Source data are provided as a Source Data file.

formulating the dynamics as a continuous-time Markov chain and finding solutions to the associated Chemical Master Equation (CME). This equation describes the probability evolution of the underlying chemical species abundances and is crucial for studying noisy intracellular reaction systems. However, solving the CME for high-dimensional systems remains a significant challenge. To address this issue, we have developed a divide-and-conquer approach called the Rao-Blackwellized CME Solver (RB-CME Solver) that combines modularization and stochastic filtering.

Specifically, the RB-CME solver works by an optimized decomposition of the system, which transforms the large-scale problem into several lower scale ones and then solves each of them using either the Monte Carlo method or a filtering approach (e.g., the filtered FSP). We showed that our method can be remarkably efficient and accurate in analyzing high-dimensional systems. Compared with the traditional Monte Carlo method, our approach is far more accurate in estimating the follower system but less accurate in estimating the leader system. However, the well-

designed system decomposition compensates for the additional costs of the filtering algorithm, making our method favorable for estimating CME solution when compared to the classical Monte Carlo method, given the same computational time constraints.

We also developed the method for the stochastic filtering problem (named the Rao-Blackwellized particle filter or the RB-PF) and showed its superior performance over conventional approaches. Specifically, the RB-PF utilizes the RB-CME solver to compute the CME in the prediction step of the filtering problem. In RB-PFs, we found that the leader system can also benefit from Rao-Blackwellization, as the conditional probability distribution of the leader interacts with the probability distribution of the follower in the correction step. More interestingly, the RB-CME solver and the RB-PF have approximately the same performance when dealing with the same system. Overall, our method is a powerful tool for solving CMEs and the associated stochastic filtering problems for high-dimensional chemical reaction systems.

Furthermore, we extended our RB-PF method for cell-specific parameter identification by including the cell-specific rate parameters as unobserved species to be inferred, in the sense of conditional probability distribution, from the measurement of the cell's time-trajectory. We successfully applied this technique to an experimental time-course data-set of transcription dynamics in yeast cells, and our results revealed significant variation in rate parameters among isogenic and identically cultured yeast cells. To validate our model inference we showed a close match between the cell-specific stationary distributions computed with our inferred model and estimated through time-averages of the measured trajectory. The parameter-heterogeneity among cells can be viewed as extrinsic noise[76] whose contribution to the overall cell-to-cell variability can be easily estimated from our cell-specific inferred models, along with the corresponding intrinsic noise that is due to the random firing of reactions. To estimate these two noise components directly from time-course experimental data, we proposed a noise decomposition approach that relies on the stability of the underlying stochastic dynamics, and does not require complex dual-reporter systems[77,79,80] that are traditionally employed to perform this noise separation. We illustrate that both the model-based and the direct approach provide comparable estimates for the two noise components. This concordance further emphasizes the validity and real-world applicability of our cell-specific inference method.

Since the CME plays a central role in the stochastic analysis of biochemical reaction systems, our method can also be generalized to consider other computational problems in biology. For instance, it is worth extending our method for the parameter sensitivity analysis and power spectrum analysis of stochastic reaction systems, which could also be computationally demanding in high-dimensional cases[12,93–95]. Also, our method might be helpful in estimating the probability of rare events, which can be difficult to achieve using conventional importance sampling approaches[96]. Computing stationary distribution is another potential application of our approach (with an idea presented in Supplementary Information, Section S9), which might improve the state-of-the-art methods (e.g.[97,98]). Moreover, the divide-and-conquer idea used in our method can also be applied to computing other equations that capture the probability evolution of stochastic systems, e.g., the Fokker-Planck equation and the associated filtering problems (see[99] for an example).

Finally, there is room for improvement of our method in terms of generality and computational efficiency. To tackle more complex systems, our method needs further development to incorporate additional biological mechanisms such as time delays, multiscale phenomena, and cell-to-cell interactions. Additionally, it is worth exploring ways to enhance the performance of the RB-CME solver, such as incorporating deep learning methods[50–52] to our divide-and-

conquer approach. Ideas for the improvement can potentially be borrowed from ref. 100, which proposed an effective deep-learning method for analyzing time delay systems.

## Methods

### Mathematical derivations
Detailed mathematical derivations and additional information are provided in Supplementary Information.

### Statistics and reproducibility
In the analysis of the experimental data, we investigated all the 130 cells that were measured on the microscope platform[90,92] over a total period of four hours. No data were excluded from the analyses of model identification and noise decomposition.

### Reporting summary
Further information on research design is available in the Nature Portfolio Reporting Summary linked to this article.

## Data availability
The mRNA measurement data in Figs. 9 and 10 have been deposited at "https://github.com/ZhouFang92/Rao-Blackwell-method-for-cell-specific-model-identification". The simulation and inference results presented in all the figures were generated by custom code, which is publicly available at the indicated GitHub repositories[101,102]. Source data are provided with this paper.

## Code availability
The code for data generation and analysis is available at the GitHub repositories: https://github.com/ZhouFang92/Rao-Blackwellized-CME-solver[101] (for the RB-CME solver) and https://github.com/ZhouFang92/Rao-Blackwell-method-for-cell-specific-model-identification[102] (for the model identification algorithm).

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

## Acknowledgements

This project has received funding from the Swiss National Science Foundation (SNSF) under grant 182653.

## Author contributions

Z.F., A.G., and M.K. designed research; Z.F. and A.G. performed research; S.K. conducted the biological experiment; Z.F., A.G., and M.K. wrote the paper.

## Competing interests

The authors declare no competing interests.
