## [Peer Review File · Nature Communications]

Advanced methods for gene network identification and noise decomposition from single-cell dataReviewer #1 (Remarks to the Author):

The well-written manuscript, titled 'Advanced Methods for Gene Network Identification and Noise Decomposition from Single-Cell Data,' introduces a more efficient computational method for addressing chemical master equations (CMEs), called the Rao-Blackwellized CME (RB-CME) solver. The manuscript illustrated that the RB-CME solver can offer a more accurate estimation of the joint distribution of species counts when compared to both the Monte-Carlo method with a fixed computational cost. Furthermore, it applied the RB-CME solver to filtering and model identification (i.e., parameter inference) problems, demonstrating its superior performance over existing methods. Finally, the method was applied to experimental data, showcasing its success in decomposing noise into extrinsic and intrinsic components.

Major

1. Estimating the stationary distribution of a system can be accomplished not just by solving the corresponding CME over an extended duration but also by identifying the steady-state solution of the CME. Can the foundational concept of the RB-CME solver be expanded to efficiently estimate the steady-state solution of a CME? Specifically, can we efficiently solve the steady-state solution by partitioning the entire system into a leader system and multiple follower systems and solving the CME independently for each follower system?

2. The advantage of the RB-CME solver, when compared to the Monte-Carlo method, is its heightened accuracy in estimating the joint distribution of species counts while keeping the computational cost constant. Notably, the new method achieved comparable accuracy to the Monte-Carlo method in estimating the marginal distributions, while excelling in providing a more precise estimate of the joint distribution. However, in a biological context, the primary interest often lies in the distribution of a specific chemical species rather than the joint distribution of multiple species. Highlighting the significance of calculating joint distributions in a biological context can further enhance the impact of the manuscript.

3. Previous studies have often simplified CME systems consisting of variables evolving on multiple timescales into a reduced system that consists only of slow variables (doi: 10.1371/journal.pcbi.1005571, 10.1371/journal.pcbi.1008952). This process involves replacing the fast variables with their stationary moments conditioned on the present slow variable states, assuming that the fast variables quickly evolve to their stochastic quasi-steady-state. It appears that this reduction method could be extended to a non-multiscale system by leveraging the idea of the RB-CME solver. Specifically, it seems possible to simplify a CME system into a reduced system consisting only of the leader system's species by replacing the follower systems' species with their moments obtained from the filtered CME. Although the reduced system could be non-Markovian, further exploration of this aspect would be intriguing in future research.

4. When dividing an entire system into a leader system and several follower systems, would making the leader system consist solely of fast (or slow) species result in a more efficient RB-CME solver?

5. The measurement noise intensity (σ) appears to have been fixed at 1 when applying the Rao-Blackwell identification algorithm to experimental data. Is it possible to infer the value of σ along with the other model parameters?

Minor

1. Page 8, Supplementary Information. Providing an intuitive explanation for each term of the filtered CME (Eq. [15]) and its specific example for a biological system would enhance readers' understanding, thereby amplifying the impact of the manuscript.

2. Page 18, Discussion section. The authors mention the potential extension of the work for time delay system incorporating deep learning method. A recent study identified that PINN can be used to estimate time delay distribution (doi.org/10.1101/2023.07.31.551393). Can this be linked with Rao-Blackwell identification?

Reviewer #2 (Remarks to the Author):

See the attached pdf file

Reviewer #2 Attachment on the following page

Review of: Advanced methods for gene network identification and noise decomposition from single-cell data

November 23, 2023

1 General Comments

This article introduces a number of related methods for the analysis of high-dimensional, stochastic mechanistic models. The article begins by a thorough assessment of the current state of the art of mechanistic modeling of biological systems within a single cell. The motivation for developing the methods in this article is the fact that high-dimensional systems are difficult to address numerically using existing methods due to the non-independence of interacting agents that are part of the system. The key innovation of these algorithms is that they decompose the system into “leader” and “follower” sub-systems, where the follower sub-systems are assumed conditionally independent given the leader system. The assumption that a system can be decomposed in this manner may not be applicable to all high-dimensional systems, but it appears to be a useful assumption in inter-cellular modeling.

We endorse various of the premises of this manuscript. There is scientific value in filtering stochastic dynamic models, yet there are outstanding questions about how to do this in practice for systems with more than a few latent variables. The authors develop new methodology for this problem, motivated by concrete scientific questions concerning single-cell molecular dynamics.

The authors point out that the particle filter is an effective way to deal with filtering for general low-dimensional systems, but that its effectiveness declines rapidly with dimension. The particle filter provides essentially an exact solution to the filtering problem, subject to Monte Carlo variability, but in higher dimensions it may be necessary to make some approximations.

The expertise of this review primarily concerns statistical methodology and data analysis, rather than single cell biology. We therefore focus on methodological issues. Our points below suggest that the new methodology may not be a substantial advance on the general inference problem within which this data analysis sits, and the authors have not placed it in that context. In the narrower sense of being an advance on analysis of this particular data type, the contribution may still be substantial.

2 Numbered points

1. An approximate word count was obtained by running the commands `pdftotext`, removing the abstract, references, all images and their captions, and other spurious symbols created by the pdf to text conversion, and then running `wc -w` on the resulting file. Doing this results in an estimated word count of 12,851 words. While this is likely an overestimate of the total word count, and the journal specifically mentions flexibility in the total word count of submitted articles, it is clear that this article is significantly longer than the suggested 5000 word limit. This may indicate that the article is outside the size and scope of this journal, but that is an editorial decision so we defer that question to the editors.
2. Frequently in the introduction, the difficulty of modeling high-dimensional systems is mentioned. We note that the dimensions considered in practice are not very large (say, a system of 7 species). This is beyond the scope of the basic particle filter, and yet contains interesting problems that are not necessarily handled well by scaleable approximations such as the ensemble Kalman filter. The authors only weakly connect to the large literature on particle filtering and its extensions to higher dimensional systems. Focusing on an application, as the authors do, can be a good way to motivate new methods. However, various existing filters could be applied to this problem, for example, the various filters compared by Ionides et al. (2023).
As a related point, in state-space modeling, a high number of dimensions could refer to either or both the number of latent states and the number of observable states. Additional clarity about the limits of practical scalability could be helpful.
3. The authors have continued a line of research on their specific scientific problem, and many of the references to particle filters correspond to work in their group (e.g., p7. references for particle filters [21,66,67]). Unless these methods are very specific to single-cell molecular dynamics, either (i) the authors methods advance the general methodological problem, or (ii) advances in the general methodological problem could contribute to this task. We think it is the authors' responsibility to justify which of these two is the case, and that better connection to the filtering literature will be required to do that.
4. A relevant recent methodology is Whitehouse et al. (2023). My guess is that this approach may be well suited to the models under investigation in this manuscript.

5. The abstract focuses on the chemical master equation (CME) solution, but the key scientific task is filtering in order to carry out parameter estimation from data. The authors point out correctly that a key step toward probabilistic filtering is to represent the one-step forecasting distribution, and this is closely related to the CME for the problem at hand. However, the other component of filtering is the data assimilation step, and that is where the difficulty arises for the particle filter in high dimensions.
6. For parameter estimation, the authors explain that they treat the parameters as follower components. However, C1 requires that “Each reaction involves a maximum of one follower sub-system (meaning that at most one follower subsystem can influence the reaction’s propensity or have its state altered by the reaction).” Does that mean that a reaction cannot have more than one parameter? Or that all parameters must be in the same sub-system, thus potentially limiting the number of parameters that can be estimated due to scalability considerations? We were somewhat confused about C1 - for example, Fig 2A seems to entail the possibility that a leader reaction will influence (or be influenced by) all three follower subsystems. Perhaps the statement of C1 could be clarified?
7. It seems to us that the authors propose a very general approach to parameter inference via a particle filter for a state space model. We can always consider the original state space model to be the leader system and each estimated parameter to be a follower subsystem. That is a simple, special case of how the authors propose to do inference. Has this approach been proposed in the extensive literature on parameter estimation via particle filters? If not, is the authors’ paper a breakthrough on this more general problem? That is for the authors to address, but we offer some conjectures.
 - (a) The authors discretize the parameter space in order to implement their method. Doing this well may be hard in practice. For example, in some of the actual data analysis in section 3B-1, the authors only consider discrete values of k_{p_1} and k_{p_2} that have very large gaps (jump size of 8 and 10 respectively). Are values $k_{p_1} = 12$ similar enough to $k_{p_1} = 8$ or $k_{p_1} = 16$ that we can ignore that possibility? Alternative available methods do not have this limitation.
 - (b) A parameter affecting a lead reaction between species A and B seems to have to be part of any sub-system joined to species A, and for similar reasons is part of any sub-system joined to B. Therefore, it seems that including this parameter requires that all other species dynamically linked to A or B (including all parameters for corresponding reaction rates) must be part of a single sub-system. Perhaps we misunderstand.
 - (c) Even if C1 does not hold exactly, assuming that it does hold in order to calculate conditional distributions may amount to a useful approximation. In particular, this means that the method does not target the exact posterior even in the limit as the number of particles tends to infinity and the discretization becomes increasingly fine. This approximation, which involves ignoring some, but not all, dependencies) is reminiscent of the approximation in the block particle filter (Rebeschini and Van Handel, 1995) which has proved useful.
8. Continuing this point, the authors claim to have made an advance on the general problem of inference for state space models, e.g., page 8, “For this reason, we require our identification algorithm to classify all the parameters as follower components, which allows their inference to be aided by a filtering approach (e.g., filtered FSP) rather than being identified purely by classical particle filtering.” Strong claims should require strong evidence, but the explanation of the inference part of this paper is brief. Most of the effort is spent explaining a factored solution to a CME. The authors are correct that this has implications for inference, along the lines of what they suggest. However, their reference on the general problem (68) is two decades out of date, in an area that has been extensively studied. We doubt that the authors really have found a breakthrough on this general problem, and we suspect that the underlying difficulties of particle depletion and exponential scaling must be hiding under the surface. For example, if there are more than a few parameters in any sub-system, the discretization on the resulting product space may have to be very sparse for FSP to operate - that is a type of curse of dimensionality.
9. Page 2, left, line 10: this statement would benefit from a citation: “This method can be computationally efficient, but its accuracy decreases as the dimension of the system increases, making it unsuitable for high-dimensional settings”.
10. Page 2, right, line 14. Missing comma in: “When the system is multiscale, i.e. it has reactions firing at different time-scales[,] this curse of dimensionality ...” Maybe break up this long sentence.
11. The rhetoric question in paragraph 9 feels out of place, and should be rephrased: “Would not it be advantageous if we could divide a high dimensional system into smaller pieces, thereby mitigating the curse of dimensionality?”
12. In paragraph 12, change “boils down” to “reduces”.
13. Last sentence in Caption of Fig. 2 should be fixed: “Since each subsystem is low dimensional by our system decomposition algorithm, our computation approach scales more favorably with the system dimension”. Perhaps: “Since our decomposition algorithm controls the size of each subsystem, our computational approach ...”
14. In table 1, various abbreviations are introduced for algorithms. The MC abbreviation is never used in the paper, as the authors have elected to use the unabbreviated version throughout the text. This entry in the table should then be removed, or the text otherwise modified to use the abbreviation.

15. Reword sentence in paragraph 3 of Section 1A: “When the molecular counts of different chemical species are independent, this sum does grow exponentially with n , as its value equals to ...” Possibly: “When the molecular counts of different chemical species are independent, this sum ~~does~~ grow[s] exponentially with n , as its value ~~equals~~ [is equal] to ...”. Maybe rephrase to clarify that the exponential growth happens with or without independence.
16. Missing punctuation in Section 1D, paragraph 1: “Also, this in turn can lead to the development of improved control strategies for the reacting process[.] Here, we apply ...”
17. First paragraph Section 1E, suggest: “This fact gives rise to another important topic in biology, i.e., model identification, which aims to develop robust and accurate math[ematical] models for biological processes from given datasets. Securing a good ~~math~~ model can provide...”
18. Section 1E focuses on model identification. Here, model identification appears to refer to the calibration of model parameters for a fixed set of mathematical equations, rather than the identification of alternate mathematical descriptions of the same system. This may be standard terminology within the modeling community of intracellular reaction systems. In general statistical terminology, this task is called parameter estimation, and “model identification” sounds more like “model selection” which is generally considered to be a different problem, concerning choices about the structure of the model. For a general purpose scientific journal, this choice of terminology should be made clear.
19. Section 1E, fifth paragraph. Suggest: “~~In addition, it is noteworthy that~~ [C]lassical particle filtering is generally ... For this reason, we require our identification algorithm to classify all of the parameters as follower components [(C4)], which allows ...”
20. In the next paragraph, suggest: “~~It is also worth noting that~~ [T]he idea of adapting the Rao-Blackwell method ... ”.
21. In paragraph 2 of section 2B, suggest: “This whole procedure took 10 graphics processing units (GPUs) [approximately] 24 hours[.]” (remove the exclamation mark). Similarly in section 2C, paragraph 2, remove the exclamation mark: “This procedure took 6.5 hours[.]”
22. In section 2C, paragraph 3: “From figure 6E, we can observe that for the follower system, the solution of the RB-PF is more smooth and accurate [than the solution of the PF, demonstrating] the advantage of RB-PF.”
23. In section 3A, you say that the numeric simulations have a uniform prior, but if these are direct simulations from a distribution then they aren’t really prior distribution. Instead, you could say: “All these parameters were in units of minute^{-1} and were simulated from uniform distributions.”
24. The simulation study only used a single draw of parameters from a uniform distribution. Would the results be significantly different if more draws were taken? What if parameters came from distributions other than uniform?
25. Paragraph 3 of section 3A states that the conditional distributions of the parameters shown in Figure 9A have a high level of confidence. This seems to be an optimistic statement, as a significant amount of mass is considerably far away from the true parameter. For example, a 95% credible interval would probably cover the range [0.2, 0.7], which is half of the range of the considered values [0, 1].
26. The manuscript has several footnotes, which should not be used: <https://www.nature.com/ncomms/submit/article>. One particular example is paragraph 4 of section 3A: “However, thanks to the stability is that the occupation time distribution ...”. This sentence doesn’t read very well, and I don’t see reason that the elaboration found in the footnote couldn’t be contained in the main text.
27. In 3A there are two numeric simulations. In the second, where only two gene states are present in the target system, generating parameter values are provided in the main text; why are they explicitly provided here and not the first simulation?
28. Last paragraph of 3A you claim the stationary distributions of the target system and the inferred models match perfectly, but this isn’t true, they just match closely.
29. In Figure 10A you only show the results for cell #78. Why this particular cell? Obviously it would be difficult to show the results for all cells, but there should at least be some justification of why the results for this cell are representative of all of the cells. Do you get worse results if you look at others?
30. In Figure 11 B: you say that all of the dots are distributed around the diagonal lines, but to me it seems that most points fall below the line, so the variance of the inferred model is consistently lower than the variance of the experiment, so the variance from the inferred models is consistently biased low.
31. The link between the Rao-Blackwell theorem and the so-called Rao-Blackwellized particle filters is not clear. This should have something to do with the relationship between the proposed algorithms and the Rao-Blackwellised particle filter of Kevin Murphy and Stuart Russell (see chapter 24 of Sequential Monte Carlo Methods in Practice, Arnaud Doucet et al).

3 Supplement

- (i) Several very similar algorithms are presented, with only a line or so different. If they could be combined into a single representation, with the few differing lines identified (e.g., formally by an if statement) that would save space and add clarity.
- (ii) In the supplement, line 81: "... can also solve the filtering problem by using the Monte Carlo for the CME step", should be "the Monte Carlo method", or otherwise remove the word "the".
- (iii) Supplement line 135: "we simplifies" should be "we simplify".
- (iv) Line 194: "when discussing other conditional probability, we always refer to such a cadlag one". This sentence could be improved. Maybe something like: "when discussing other conditional probability, we are referring to a distribution that is cadlag".
- (v) Line 195: "we lists" should be "we list".
- (vi) Line 297: "we particular choose" could be "we specifically choose", or the word particular could be removed altogether.
- (vii) Line 364, there is a typo, $h_j(\tilde{x}, z_1, \dots, z_1)$ should be $h_j(\tilde{x}, z_1, \dots, z_l)$
- (viii) Line 377, "We prove the rest result", remove the word "rest".
- (ix) Line 396: "Moreover, by math induction", formally, "Moreover, by the principle of mathematical induction,"
- (x) Line 7 of Algorithm 6: The notation here could be improved. In particular, the samples $\tilde{z}_1(t_i), \dots, \tilde{z}_l(t_i)$ are unique for each particle. This is reflected for the "leader particles": $\tilde{x}_j(t_i)$, so this should also be reflected in the follower particles, maybe: $\tilde{z}_{l,j}(t_i)$, or $\tilde{z}_l^j(t_i)$
- (xi) Line 469: "A rigorous analysis for that requires the techniques in (4), which is somehow complicated.". Saying that it is somehow complicated feels a bit informal, maybe replace with "which could be the topic of future work", or "which is outside the scope of this analysis", or equivalent.

4 References

- Ionides, E. L., Asfaw, K., Park, J., & King, A. A. (2023). Bagged filters for partially observed interacting systems. *Journal of the American Statistical Association*, 118(542), 1078-1089.
- Rebeschini, P., & Van Handel, R. (2015). Can local particle filters beat the curse of dimensionality? *Annals of Applied Probability*, 25(5), 2809-2866.
- Whitehouse, M., Whiteley, N., & Rimella, L. (2023). Consistent and fast inference in compartmental models of epidemics using Poisson Approximate Likelihoods. *Journal of the Royal Statistical Society Series B: Statistical Methodology*, 85(4), 1173-1203.

Reviewer #3 (Remarks to the Author):

Please see the attached PDF file for the review.

I co-reviewed this manuscript with one of the reviewers who provided the listed reports as part of the Nature Communications initiative to facilitate training in peer review and appropriate recognition for co-reviewers.

Authors' response to the reviewers and list of actions taken

Zhou Fang, Ankit Gupta, Sant Kumar, Mustafa Khammash

March 6, 2024

We thank the referees for the positive recommendation of our manuscript. Responding to the referees' comments has helped us considerably improve the quality of the manuscript. The major changes in the revision were:

- Addition of identification results for the simulated and experimental data to illustrate the broad applicability of our method in single-cell analysis problems,
- The discussion and literature review sections were considerably expanded to better establish the connection between our method and existing approaches (i.e., the state-of-the-art filtering approaches and the time-scale separation method for chemical reaction networks).

We have addressed all of the issues raised by the referees in the revised manuscript. Below, we list the referees' comments, followed by our detailed response and actions taken. For the convenience of the editor and the referees, we have highlighted in blue the parts of the manuscript that have been changed.

Response to Referee 1

1. *The well-written manuscript, titled 'Advanced Methods for Gene Network Identification and Noise Decomposition from Single-Cell Data,' introduces a more efficient computational method for addressing chemical master equations (CMEs), called the Rao-Blackwellized CME (RB-CME) solver. The manuscript illustrated that the RB-CME solver can offer a more accurate estimation of the joint distribution of species counts when compared to both the Monte-Carlo method with a fixed computational cost. Furthermore, it applied the RB-CME solver to filtering and model identification (i.e., parameter inference) problems, demonstrating its superior performance over existing methods. Finally, the method was applied to experimental data, showcasing its success in decomposing noise into extrinsic and intrinsic components.*

Authors' response: Many thanks for your detailed summary and positive comments on our paper, which we find very encouraging. We have carefully considered each of your detailed comments. Our responses and revisions to each point are provided below.

2. *Estimating the stationary distribution of a system can be accomplished not just by solving the corresponding CME over an extended duration but also by identifying the steady-state solution of the CME. Can the foundational concept of the RB-CME solver be expanded to efficiently estimate the steady-state solution of a CME? Specifically, can we efficiently solve the steady-state solution by partitioning the entire system into a leader system and multiple follower systems and solving the CME independently for each follower system?*

Authors' response: This is an important point concerning the extension of our approach for computing the steady-state solution of the CME. Essentially, the conditional independence of the follower subsystems is established based on the trajectory (rather than the current state) of the leader state. Consequently, in the general case, it is challenging to directly apply our method to solve the steady-state CME. When the follower systems are fast, the conditional distribution of follower species can be approximated by the quasi-stationary distribution, and, therefore, we can obtain the conditional independence of the follower subsystems given only the current state of the leader system rather than its full historical trajectory. In this case, our approach is similar to the time-scale separation method and can be directly applicable to the

steady-state CME by solving several subproblems independently. This idea is in line with the time-scale separation methods [Gou05, CGP05, EVE05, KK13, KS17, SHK21] but applied to the steady-state CME, which can be computationally more efficient than the conventional finite-state projection method. We have provided a comprehensive discussion about the connection between the time-scale separation method and our RB-CME solver in SI Appendix, Section S3.E.

Even though the direct application of the RB-CME solver to the steady-state CME is challenging, our method can also be applied to solve this problem in an indirect way. As mentioned by the reviewer, we can tackle this problem by solving the corresponding CME over an extended duration. Also, we can compute the steady-state solution by combining the ergodicity (if it exists) with the RB-CME solver. Specifically, when the system is ergodic, the stationary distribution can be approximated by

$$\mathbb{P}_{st}(x) \approx \underbrace{\frac{1}{T} \int_0^T \mathbb{1}(X(t) = x) dt}_{\hat{\mathbb{P}}_{er}(x)}$$

for a large T . When combining it with our RB-CME solver, we can approximate the stationary distribution by

$$\mathbb{P}_{st}(x) \approx \underbrace{\frac{1}{T} \int_0^T \mathbb{E} \left[\mathbb{1}(X(t) = x) \middle| \tilde{X}(s), 0 \leq s \leq t \right] dt}_{\hat{\mathbb{P}}_{RB}(x)}$$

where $\tilde{X}(t)$ is the leader system, and the conditional distribution is computed by the RB-CME solver. It can be easily shown that

$$\begin{aligned} \text{Var} \left(\hat{\mathbb{P}}_{er}(x) \right) &= \mathbb{E} \left[\left(\hat{\mathbb{P}}_{er}(x) - \hat{\mathbb{P}}_{RB}(x) \right)^2 \right] + \mathbb{E} \left[\left(\hat{\mathbb{P}}_{RB}(x) - \frac{1}{T} \int_0^T \mathbb{P}(X(t) = x) dt \right)^2 \right] \\ &\quad + 2\mathbb{E} \left[\left(\hat{\mathbb{P}}_{er}(x) - \hat{\mathbb{P}}_{RB}(x) \right) \left(\hat{\mathbb{P}}_{RB}(x) - \frac{1}{T} \int_0^T \mathbb{P}(X(t) = x) dt \right) \right] \end{aligned}$$

Note that $\mathbb{E} \left[\left(\hat{\mathbb{P}}_{RB}(x) - \frac{1}{T} \int_0^T \mathbb{P}(X(t) = x) dt \right)^2 \right]$ is the variance of $\hat{\mathbb{P}}_{RB}(x)$, and the last term satisfies

$$\begin{aligned} &\mathbb{E} \left[\left(\hat{\mathbb{P}}_{er}(x) - \hat{\mathbb{P}}_{RB}(x) \right) \left(\hat{\mathbb{P}}_{RB}(x) - \frac{1}{T} \int_0^T \mathbb{P}(X(t) = x) dt \right) \right] \\ &= \mathbb{E} \left[\mathbb{E} \left[\left(\hat{\mathbb{P}}_{er}(x) - \hat{\mathbb{P}}_{RB}(x) \right) \left(\hat{\mathbb{P}}_{RB}(x) - \frac{1}{T} \int_0^T \mathbb{P}(X(t) = x) dt \right) \middle| \tilde{X}(s), 0 \leq s \leq T \right] \right] \\ &= \mathbb{E} \left[\mathbb{E} \left[\hat{\mathbb{P}}_{er}(x) - \hat{\mathbb{P}}_{RB}(x) \middle| \tilde{X}(s), 0 \leq s \leq T \right] \left(\hat{\mathbb{P}}_{RB}(x) - \frac{1}{T} \int_0^T \mathbb{P}(X(t) = x) dt \right) \right] \\ &= \mathbb{E} \left[0 \times \left(\hat{\mathbb{P}}_{RB}(x) - \frac{1}{T} \int_0^T \mathbb{P}(X(t) = x) dt \right) \right] \\ &= 0. \end{aligned}$$

Therefore, we can conclude that

$$\text{Var} \left(\hat{\mathbb{P}}_{er}(x) \right) \geq \text{Var} \left(\hat{\mathbb{P}}_{RB}(x) \right)$$

which means that the new approximation method based on Rao-Blackwellization is no less accurate than the former one. We leave a systematic exploration of this idea for future work. We also mentioned this potential future work in the discussion section (please see the second last paragraph in the discussion section).

3. *The advantage of the RB-CME solver, when compared to the Monte-Carlo method, is its heightened accuracy in estimating the joint distribution of species counts while keeping the computational cost constant. Notably, the new method achieved comparable accuracy to the Monte-Carlo method in estimating the marginal distributions, while excelling in providing a more precise estimate of the joint distribution. However, in a biological context, the primary interest often lies in the distribution of a specific chemical species rather than the joint distribution of multiple species. Highlighting the significance of calculating joint distributions in a biological context can further enhance the impact of the manuscript.*

Authors' response: Thanks for pointing this out. In many cases, biologists not only focus on the marginal distribution of a specific chemical species but also the joint distribution of several considered species. The joint distribution can offer insights into the interaction mechanisms between different species and their combined effects. This is particularly important for systems whose functionality is a result of the combined effects of many species, e.g., the repressilator and genetic toggle switch. In the revised manuscript, we have emphasized this point in the introduction section (see the highlighted text in the third paragraph of the introduction section) and in the numerical example of the repressilator (see the second paragraph of Section 2.B).

4. *Previous studies have often simplified CME systems consisting of variables evolving on multiple timescales into a reduced system that consists only of slow variables (doi: 10.1371/journal.pcbi.1005571, 10.1371/journal.pcbi.1008952). This process involves replacing the fast variables with their stationary moments conditioned on the present slow variable states, assuming that the fast variables quickly evolve to their stochastic quasi-steady-state. It appears that this reduction method could be extended to a non-multiscale system by leveraging the idea of the RB-CME solver. Specifically, it seems possible to simplify a CME system into a reduced system consisting only of the leader system's species by replacing the follower systems' species with their moments obtained from the filtered CME. Although the reduced system could be non-Markovian, further exploration of this aspect would be intriguing in future research.*

Authors' response: This is a very important point. When the filtered CME is approximated by moment closure, and the leader system is simulated together with the moments of the follower species, the proposed Rao-Blackwell framework leads to a model reduction method. This aligns with the uncoupled simulation method introduced in the references [58, 59]. In the original version of this paper, we have reviewed this uncoupled simulation method and drawn connections between it and our method. In the revised manuscript, we rephrased some sentences to further enhance the relevant discussion. Please check these discussions in the second last paragraph of the right column on Page 2 and the first paragraph of the right column on Page 3.

Even though many papers have been devoted to the uncoupled simulation algorithms, many open questions still exist. For instance, it is still unclear how to incorporate such a method into time-delay systems, which moment-closure approach can work best for such a reduction method, how to find the best decomposition for the reduction purpose, and how to quantify the error between the approximated solution and the exact one. Definitely, many problems can be explored in future work.

5. *When dividing an entire system into a leader system and several follower systems, would making the leader system consist solely of fast (or slow) species result in a more efficient RB-CME solver?*

Authors' response: This is an important point concerning the connection between the time-scale separation approaches and our method. First, the time-scale separation method and the RB-CME solver share many similarities. Both methods used the Monte-Carlo samples to estimate a part of the species (the slow species or the leader species) and combine these estimates with conditional distributions (by the quasi-stationary assumption or filtered FSP) to estimate the remaining species. However, they are developed based on different motivations. The time-scale separation approach is designed to simplify the analysis and simulation of multi-scale systems. In contrast, the RB-CME solver aims to improve the estimation methods for systems whose species are at similar time scales. The current system decomposition algorithm is also developed based on the same-time-scale setting.

Coming back to the problem of applying the RB-CME solver to multiscale systems, we suggest to first decompose the system based on the time scales and then apply the RB-CME solver to the reduced system. When the species are at the same time scale but exhibit non-negligible fast-slow behaviors, the system decomposition algorithm needs to be re-designed accordingly. Particularly, we think it would be better to classify all the slow species as the leader species and the faster species as follower species, as this would lead to more reduction in the estimation error. A more detailed discussion about this point is now provided in SI Appendix, Section S3.E. In general, extending the RB-CME solver to (relatively) multi-scale systems is a challenging problem; it also needs to consider the size of the leader system, the stiffness and the size of the filtered CME for each follower subsystem. We leave a systematic exploration of this problem for future work.

In the revised paper, we have provided a comprehensive discussion about the above points in SI Appendix, Section S3.E. Also, in the main text, we mentioned the existence of this discussion (see the first paragraph on the right column of Page 6).

6. *The measurement noise intensity (σ) appears to have been fixed at 1 when applying the Rao-Blackwell identification algorithm to experimental data. Is it possible to infer the value of σ along with the other model parameters?*

Authors' response: Thanks for this comment. Yes, our algorithm is applicable when the measurement noise intensity σ is unknown. In this situation, we can view σ as an additional unknown parameter and infer it simultaneously with other parameters. To illustrate this point, we performed numerical experiments to test the performance of our Rao-Blackwell method in this situation. The numerical results are shown in Section S7.D and Figure S2 (SI Appendix) in the revised manuscript. Also, we briefly discussed this point in the last paragraph of Section 3A on Page 14. The numerical results indicate that our method can accurately identify the measurement noise intensity σ and other parameters in the presence of uncertainty in σ . Moreover, our method performs consistently well when the real value of σ is chosen differently. Please see these new results and additional details in the revised manuscript.

7. *Page 8, Supplementary Information. Providing an intuitive explanation for each term of the filtered CME (Eq. [15]) and its specific example for a biological system would enhance readers' understanding, thereby amplifying the impact of the manuscript.*

Authors' response: Following this suggestion, we have added a discussion below Eq. [15] in the revised supplementary information to explain each term in the filtered CME. Also, we have added one simple gene-expression example to enhance readers' understanding of the filtered CME. These can be found on Page 8 of the Supplementary Information.

8. *Page 18, Discussion section. The authors mention the potential extension of the work for time delay system incorporating deep learning method. A recent study identified that PINN can be used to estimate time delay distribution (doi.org/10.1101/2023.07.31.551393). Can this be linked with Rao-Blackwell identification?*

Authors' response: Thanks for suggesting this paper! We also noticed this paper recently and found it interesting. The deep learning method developed in that paper is very effective for analyzing the time delay distribution. We believe this deep learning method can be linked with our Rao-Blackwell identification method. For instance, in a time-delay system, we can classify all the species as leader components, and the time delay distribution/coefficient as the follower part. Then, the Rao-Blackwell method needs to infer the time delay distribution/coefficient from the trajectory of the leader components (the chemical species), which can be effectively solved by the deep learning approach in the suggested paper. This is a preliminary idea, and we believe a more sophisticated method combining the Rao-Blackwell method and the deep learning approach could be developed with further research, possibly in future work. We have briefly mentioned this point at the end of the discussion section.

Response to Referee 2 and 3

1. *This article introduces a number of related methods for the analysis of high-dimensional, stochastic mechanistic models. The article begins by a thorough assessment of the current*

state of the art of mechanistic modeling of biological systems within a single cell. The motivation for developing the methods in this article is the fact that high-dimensional systems are difficult to address numerically using existing methods due to the non-independence of interacting agents that are part of the system. The key innovation of these algorithms is that they decompose the system into “leader” and “follower” sub-systems, where the follower sub-systems are assumed conditionally independent given the leader system. The assumption that a system can be decomposed in this manner may not be applicable to all high-dimensional systems, but it appears to be a useful assumption in inter-cellular modeling.

We endorse various of the premises of this manuscript. There is scientific value in filtering stochastic dynamic models, yet there are outstanding questions about how to do this in practice for systems with more than a few latent variables. The authors develop new methodology for this problem, motivated by concrete scientific questions concerning single-cell molecular dynamics.

The authors point out that the particle filter is an effective way to deal with filtering for general low-dimensional systems, but that its effectiveness declines rapidly with dimension. The particle filter provides essentially an exact solution to the filtering problem, subject to Monte Carlo variability, but in higher dimensions it may be necessary to make some approximations.

The expertise of this review primarily concerns statistical methodology and data analysis, rather than single cell biology. We therefore focus on methodological issues. Our points below suggest that the new methodology may not be a substantial advance on the general inference problem within which this data analysis sits, and the authors have not placed it in that context. In the narrower sense of being an advance on analysis of this particular data type, the contribution may still be substantial.

Authors’ response: We sincerely thank the referees for the in-depth review and many positive comments about our paper. We have revised the paper based on the feedback. Here, we would like to respond to your comments regarding the contribution of this work.

This paper aims to develop effective methods for analyzing high-dimensional intracellular reaction processes. As the referees pointed out, such problems are challenging even though many methods (e.g., particle filtering) exist. Since our approach performs nicely for solving CMEs (which are central to biological system analysis), the proposed methods can work effectively for broad biological problems. We believe that the extensive numerical and experimental examples (together with the theoretical results) are sufficient to show the good performance of our approach for the targeted problems. In the revised manuscript, we have added

- (a) more identification results (based on both simulation and experimental data) to illustrate the effectiveness of our approach in single-cell identification problems (see the responses to Comment 8(a) and 25)
- (b) more references on stochastic filtering to justify the design of our method and its connection to existing approaches (see the responses to Comments 3, 4, 5, 6, and 8)

This paper demonstrated the effectiveness of the proposed Rao-Blackwell method for single-cell filtering and parameter inference problems. However, the exploration of this method’s effectiveness for general mechanistic models beyond single-cell systems falls outside the scope of this study. It requires more in-depth theoretical and numerical analysis of the approach in a general setting other than biology. We leave it for future work. Our recent arXiv preprint [FGK23b] has provided some preliminary results for this problem.

2. *An approximate word count was obtained by running the commands `pdftotext`, removing the abstract, references, all images and their captions, and other spurious symbols created by the pdf to text conversion, and then running `wc -w` on the resulting file. Doing this results in an estimated word count of 12,851 words. While this is likely an overestimate of the total word count, and the journal specifically mentions flexibility in the total word count of submitted articles, it is clear that this article is significantly longer than the suggested 5000 word limit. This may indicate that the article is outside the size and scope of this journal, but that is an editorial decision so we defer that question to the editors.*

Authors’ response: Thanks for pointing this out. We are open to make necessary revisions to comply with the journal’s requirements, if our paper is accepted for publication. Should

the editor request it, we would be willing to shorten the paper by relocating some of the more detailed mathematical derivations to the method section and shifting several numerical examples to the supplementary material.

3. *Frequently in the introduction, the difficulty of modeling high-dimensional systems is mentioned. We note that the dimensions considered in practice are not very large (say, a system of 7 species). This is beyond the scope of the basic particle filter, and yet contains interesting problems that are not necessarily handled well by scaleable approximations such as the ensemble Kalman filter. The authors only weakly connect to the large literature on particle filtering and its extensions to higher dimensional systems. Focusing on an application, as the authors do, can be a good way to motivate new methods. However, various existing filters could be applied to this problem, for example, the various filters compared by Ionides et al. (2023).*

As a related point, in state-space modeling, a high number of dimensions could refer to either or both the number of latent states and the number of observable states. Additional clarity about the limits of practical scalability could be helpful.

Authors' response: Thanks for this comment and for suggesting the paper by Ionides et al. (2023). In the revised manuscript, we have added one paragraph to review the literature on stochastic filtering, its applications to biology, and the remaining challenges (see the last paragraph in the right column on Page 2). In this paragraph, we also pointed out that the challenges in filtering problems can come from the high dimensionality of both the latent and the observed states. There, we also reviewed the bagged filter (from the suggested paper) and the well-known block particle filter, both of which work well for systems with a large number of observation channels. In addition, we also added a literature review on the joint estimation of system state and parameters in Section 3.E to support our discussion (see the last paragraph in the left column on Page 8).

This paper focuses on the analysis of biological problems specifically. Even though the systems considered in the filtering sections contain 6 or 10 hidden variables (including both species and parameters), which is not very large compared to the ones investigated in other scientific domains, these biological systems are quite high-dimensional for stochastic modeling in systems and synthetic biology. Also, due to the limited availability of distinguishable reporters, single-cell filtering problems typically have few observation channels. Therefore, this paper focuses on high-dimensional biochemical reaction systems with few observation channels. We have now clearly stated this information in the last paragraph on Page 2.

4. *The authors have continued a line of research on their specific scientific problem, and many of the references to particle filters correspond to work in their group (e.g., p7. references for particle filters [21,66,67]). Unless these methods are very specific to single-cell molecular dynamics, either (i) the authors methods advance the general methodological problem, or (ii) advances in the general methodological problem could contribute to this task. We think it is the authors' responsibility to justify which of these two is the case, and that better connection to the filtering literature will be required to do that.*

Authors' response: Thanks for pointing this out. The methods in these references are very specific to single-cell filtering problems. These methods extend the classical particle filtering method to cope with challenges in single-cell filtering problems, e.g., the multi-scale property. In the revised manuscript, we have pointed out that these references are specific to single-cell problems and additionally cited two other review papers on particle filtering here. Please see the changes in the first paragraph in the right column of Page 7.

Moreover, in the introduction section, we have added one more paragraph to review the literature on stochastic filtering theory and its applications to biology (see the last paragraph in the right column on Page 2). In section 3.E, we also added a literature review on the joint estimation of system state and parameters to support our discussion (see the last paragraph in the left column of Page 8).

5. *A relevant recent methodology is Whitehouse et al. (2023). My guess is that this approach may be well suited to the models under investigation in this manuscript.*

Authors' response: Thanks for suggesting this paper. This paper proposed a parametric method by Poisson approximation to solve the filtering problems for epidemics. In the revised paper,

we have properly cited this paper together with many other relevant ones. Please see the reference [64] in the revised manuscript and the last paragraph on Page 2. In principle, this approach in Whitehouse et al. (2023) can also be applied to single-cell filtering problems. However, like other parametric methods (including the moment-closure approach and linear noise approximation approach that we have cited in our paper), this Poisson approximation method can only work well when the actual probability distribution is exactly or closely aligned with a Poisson distribution. This restriction greatly affects the applicability of this Poisson approximation method to generic biological systems, whose probability distributions often deviate from a Poisson distribution. We have mentioned this point when citing this paper.

6. *The abstract focuses on the chemical master equation (CME) solution, but the key scientific task is filtering in order to carry out parameter estimation from data. The authors point out correctly that a key step toward probabilistic filtering is to represent the one-step forecasting distribution, and this is closely related to the CME for the problem at hand. However, the other component of filtering is the data assimilation step, and that is where the difficulty arises for the particle filter in high dimensions.*

Authors' response: Thanks for this comment. You are absolutely right that the large number of observation channels can cause more challenges in stochastic filtering. In the revised manuscript, we have explicitly pointed this out and discussed several methods (e.g., the block particle filter) capable of tackling this challenge. Please see the last paragraph on Page 2.

In single-cell filtering problems, the system typically involves very few observation channels due to the limited availability of spectrally orthogonal fluorescent reporters. Therefore, this paper only considers high-dimensional bio-chemical reaction systems with few observation channels. In the revised manuscript, we have also explicitly pointed this out in the last paragraph on Page 2.

7. *For parameter estimation, the authors explain that they treat the parameters as follower components. However, C1 requires that "Each reaction involves a maximum of one follower subsystem (meaning that at most one follower subsystem can influence the reaction's propensity or have its state altered by the reaction)." Does that mean that a reaction cannot have more than one parameter? Or that all parameters must be in the same sub-system, thus potentially limiting the number of parameters that can be estimated due to scalability considerations? We were somewhat confused about C1 - for example, Fig 2A seems to entail the possibility that a leader reaction will influence (or be influenced by) all three follower subsystems. Perhaps the statement of C1 could be clarified?*

Authors' response: This is an important question. Basically, C1 means that for each reaction in Eq. [1], the parameters and follower species influencing the propensity $\lambda_j(\cdot)$, as well as the follower species affected by the reaction, should be put in the same follower subsystem. According to C1, each reaction can have more than one parameter, and its parameters will be put into the same follower subsystem. However, parameters in different reactions can be put into different follower subsystems to improve the scalability of the overall procedure (see Fig. 8.B as an example).

In Fig 2.A, each arrow represents a conversion reaction, and its propensity only depends on the reactant (the source node of the arrow). Here, no reaction involves more than one follower subsystem. The reactions converting the leader species to each other do not involve (in the way we defined) any follower subsystem. Although one leader species interacts with more than one follower subsystem, it does not violate C1. Notably, C1 does not set restrictions on leader species. Additionally, our classification only differentiates between species as either leader or follower species, but it does not categorize the reactions in this manner.

This is also related to the issue stated in Comment 8(b). We think there you meant the reaction $A \xrightarrow{k_1} B$ where A and B are both leader species. For this reaction, only parameter k_1 affects the propensity, and no follower species influences the propensity or has its state altered after the firing of this reaction. Then, according to C1–C4, the parameter k_1 can form an individual follower subsystem, if k_1 does not affect other reactions. So, k_1 is not necessarily combined with other follower subsystems influencing or influenced by A and B .

In the revised manuscript, we rephrase C1 for better clarity: “ Each reaction in **Eq. [1]** involves a maximum of one follower sub-system (meaning that at most one follower subsystem can influence **this** reaction’s propensity or have its state altered by **this** reaction).” We hope this rephrasing and explanation resolve any confusion.

8. *It seems to us that the authors propose a very general approach to parameter inference via a particle filter for a state space model. We can always consider the original state space model to be the leader system and each estimated parameter to be a follower subsystem. That is a simple, special case of how the authors propose to do inference. Has this approach been proposed in the extensive literature on parameter estimation via particle filters? If not, is the authors’ paper a breakthrough on this more general problem? That is for the authors to address, but we offer some conjectures.*
- (a) *The authors discretize the parameter space in order to implement their method. Doing this well may be hard in practice. For example, in some of the actual data analysis in section 3B-1, the authors only consider discrete values of k_{p_1} and k_{p_2} that have very large gaps (jump size of 8 and 10 respectively). Are values $k_{p_1} = 12$ similar enough to $k_{p_1} = 8$ or $k_{p_1} = 16$ that we can ignore that possibility? Alternative available methods do not have this limitation.*
 - (b) *A parameter affecting a lead reaction between species A and B seems to have to be part of any sub-system joined to species A, and for similar reasons is part of any sub-system joined to B. Therefore, it seems that including this parameter requires that all other species dynamically linked to A or B (including all parameters for corresponding reaction rates) must be part of a single sub-system. Perhaps we misunderstand.*
 - (c) *Even if C1 does not hold exactly, assuming that it does hold in order to calculate conditional distributions may amount to a useful approximation. In particular, this means that the method does not target the exact posterior even in the limit as the number of particles tends to infinity and the discretization becomes increasingly fine. This approximation, which involves ignoring some, but not all, dependencies) is reminiscent of the approximation in the block particle filter (Rebeschini and Van Handel, 1995) which has proved useful.*

Authors’ response: Thanks for this comment. The idea of adapting the Rao-Blackwell method for parameter identification is relatively unexplored, even though some contributions in this direction exist. It’s correct that one can consider all the system states to be the leader system and parameters to be follower systems. This idea has been investigated in pioneering literature [ZUP⁺14] for biological systems where each propensity is linearly affected by one associated parameter. In the original version of our paper, we have already spent one paragraph reviewing this literature; now, we have highlighted this paragraph in the revised manuscript (see the second paragraph in the right column on Page 8). In this paragraph, we have also pointed out the significant differences between our approach and the existing one. First, the existing method in [ZUP⁺14] requires each parameter to linearly affect one associated propensity, limiting its applicability to many real-world biological problems whose parameters jointly and non-linearly affect the propensities (e.g., Michaelis-Menten kinetics and Hill-type dynamics). In contrast, our approach is not constrained by the type of kinetics and, therefore, has broader applicability. Second, our approach can classify some species as follower components, allowing more accurate estimates of both species and parameters. Please see these discussions in the second paragraph of the right column on Page 8.

The focus of this paper is to solve the forward and reverse problems in single-cell biology. We believe that the extensive numerical and experimental examples (together with some theoretical results) are sufficient to show the good performance of our approach for the targeted problems. We also noticed that the proposed idea can be generalized to solve parameter inference problems in more general settings. However, since this is not the scope of this paper, we leave it for future work. We also mentioned this in the discussion session; see the last sentence of the second paragraph in the right column on Page 18. Generally speaking, we believe the proposed idea of adapting the Rao-Blackwell method for parameter inference will be effective in other applications, and our paper hence represents a major advance on this more general problem. Our recent arXiv preprint [FGK23b] showed that this idea can be extended

to the parameter inference problems in SDEs; in one numerical example of the Lorenz-63 model, it outperforms existing approaches by a large margin. However, more careful and detailed investigations are required to explore the full potential of this idea in more general settings, so we do not make any strong claims regarding the versatility of this approach in this paper.

Our responses to the concerns and suggestions in (a), (b), and (c) are as follows.

Response to (a): Thanks for this point. Indeed, in the experimental example, we had a relatively coarse grid for the parameter spaces. However, the discretization sizes (8 and 10) are actually not very large compared to the parameter intervals $[0, 80]$ and $[20, 120]$, and they are sufficient to allow for accurate estimates of the parameters (evident by results we presented in that section). Of course, we acknowledge that scientists would like higher resolution estimates. To this end, we further proposed an improved RB-PF based on grid refinement in the revised manuscript. The key idea is that we don't need such a big parameter region (for the prior distribution) if we have more accurate information about the parameter values. Fortunately, we can gain such information after applying the inference algorithm based on the coarse grid scheme. Following this idea, we can shrink the parameter space based on the inference result under the relatively coarse grid scheme, set a finer discretization scheme, and reapply the inference algorithm. This procedure can be repeated many times until a good resolution of the parameter space is reached. In Section S7.E of the supplementary material, we introduced this grid refinement strategy and applied it to the experimental data. The result shows that this strategy can result in reduced discretization sizes (to 2 for k_{p_1} , and 4 for k_{p_2}) and provide more accurate estimates of the parameters. Please see the discussions and results in Section S7.E and Figure S3 in the supplementary material.

Response to (b): We think that you meant the reaction $A \xrightarrow{k_1} B$ where A and B are both leader species. For this reaction, only parameter k_1 affects the propensity, and no follower species influences the propensity or has its state altered after the firing of this reaction. Then, according to C1–C4, the parameter k_1 can form an individual follower subsystem, if k_1 does not affect other reactions. So, k_1 is not necessarily combined with other follower subsystems influencing or influenced by A and B .

Response to (c): Thanks for this suggestion. This paper only focuses on the decomposition based on conditional independence and doesn't consider the situation you suggested. The idea of applying the block particle filter to approximate the solution of the filtered CME is interesting and potentially very useful. We leave it for future work.

9. *Continuing this point, the authors claim to have made an advance on the general problem of inference for state space models, e.g., page 8, "For this reason, we require our identification algorithm to classify all the parameters as follower components, which allows their inference to be aided by a filtering approach (e.g., filtered FSP) rather than being identified purely by classical particle filtering." Strong claims should require strong evidence, but the explanation of the inference part of this paper is brief. Most of the effort is spent explaining a factored solution to a CME. The authors are correct that this has implications for inference, along the lines of what they suggest. However, their reference on the general problem (68) is two decades out of date, in an area that has been extensively studied. We doubt that the authors really have found a breakthrough on this general problem, and we suspect that the underlying difficulties of particle depletion and exponential scaling must be hiding under the surface. For example, if there are more than a few parameters in any sub-system, the discretization on the resulting product space may have to be very sparse for FSP to operate - that is a type of curse of dimensionality.*

Authors' response: Thanks for this comment. In the revised manuscript, we have added a literature review on the joint state-parameter estimation in Section 3.E to support our method design (see the last paragraph in the left column on Page 8). In the past two decades, several types of approaches have been introduced to mitigate the limitation of particle filters in estimating static variables. They include the resample-move method [GB01, BG01], regularized particle filter [LW01, OM00, FGK23a], and nested filters [CM18, PVM21], among which several methods were proposed quite recently. All these methods address the issue of sample degeneracy (in classical particle filters) by introducing artificial noise to the static variables. However, this artificial noise also introduces a bias to the estimate. Consequently,

these methods require a proper selection of the artificial noise intensity to balance mitigating sample degeneracy against minimizing additional bias, which is very challenging (if not impossible). To conclude, using pure Monte-Carlo samples to estimate static variables can be ineffective. Based on this observation, we designed the method to classify all the parameters as follower components to avoid the aforementioned problems. Please have a look at these discussions in the last paragraph in the left column on Page 8.

We also appreciate you pointing out the potential limitation of our method. You are right that if a few parameters are in one subsystem, then the discretization can be a bit sparse, like what we had in the experimental example. In the revised manuscript, we proposed a grid refinement strategy to gradually refine the grid and obtain higher resolution estimates. We briefly introduced this strategy in the response to comment 8(a). More details about this refinement method and its application to the experimental data can be found in Section S7.E of the supplementary material.

We acknowledge that the current method based on the filtered FSP cannot deal with the situation when a subsystem involves more than a few parameters. However, this issue can be mitigated by developing an accurate and more efficient method to solve the filtered CME. Deep learning is a potential solution. In the paper, we have listed this as future work; please see the last paragraph in the right column on Page 18.

10. *Page 2, left, line 10: this statement would benefit from a citation: “This method can be computationally efficient, but its accuracy decreases as the dimension of the system increases, making it unsuitable for high-dimensional settings”.*

Authors’ response: In the revised manuscript, we have referred the readers to the discussions in sections 2 and S2.A for further reading. Please check the last paragraph in the left column on Page 2.

11. *Page 2, right, line 14. Missing comma in: “When the system is multiscale, i.e. it has reactions firing at different time-scales[,] this curse of dimensionality ...” Maybe break up this long sentence.*

Authors’ response: We have split the sentence into two sentences. Please see the third paragraph in the right column on Page 2. To prevent visual overload and ensure ease of reading, we didn’t highlight this change in the revised manuscript.

12. *The rhetoric question in paragraph 9 feels out of place, and should be rephrased: “Would not it be advantageous if we could divide a high dimensional system into smaller pieces, thereby mitigating the curse of dimensionality?”*

Authors’ response: In the revised manuscript, we rephrased this sentence to a declarative sentence. Please see the change in the second paragraph of the left column on Page 3.

13. *In paragraph 12, change “boils down” to “reduces”.*

Authors’ response: In the revised manuscript, we have rephrased the sentence as “..., our method is equivalent to the method ...”. Please see the first paragraph in the right column on Page 3.

14. *Last sentence in Caption of Fig. 2 should be fixed: “Since each subsystem is low dimensional by our system decomposition algorithm, our computation approach scales more favorably with the system dimension”. Perhaps: “Since our decomposition algorithm controls the size of each subsystem, our computational approach ...”*

Authors’ response: Please see the revision in the caption of Fig. 2.

15. *In table 1, various abbreviations are introduced for algorithms. The MC abbreviation is never used in the paper, as the authors have elected to use the unabbreviated version throughout the text. This entry in the table should then be removed, or the text otherwise modified to use the abbreviation.*

Authors’ response: Thanks for pointing this out. We have removed it from the table.

16. *Reword sentence in paragraph 3 of Section 1A: “When the molecular counts of different chemical species are independent, this sum does grow exponentially with n , as its value equals to*

...” Possibly: “When the molecular counts of different chemical species are independent, this sum does grow[s] exponentially with n , as its value equals [is equal] to ...”. Maybe rephrase to clarify that the exponential growth happens with or without independence.

Authors’ response: Please see the changes below Eq. [4] on Page 4.

17. *Missing punctuation in Section 1D, paragraph 1: “Also, this in turn can lead to the development of improved control strategies for the reacting process[.] Here, we apply ...”*

Authors’ response: Thanks for pointing this out. We have added a period there in the revised manuscript.

18. *First paragraph Section 1E, suggest: “This fact gives rise to another important topic in biology, i.e., model identification, which aims to develop robust and accurate math[ematical] models for biological processes from given datasets. Securing a good ~~math~~ model can provide...”*

Authors’ response: We have corrected it in the revised manuscript. Please have a look at the first paragraph of Section 1.E. To prevent visual overload and ensure ease of reading, we didn’t highlight this change in the revised manuscript.

19. *Section 1E focuses on model identification. Here, model identification appears to refer to the calibration of model parameters for a fixed set of mathematical equations, rather than the identification of alternate mathematical descriptions of the same system. This may be standard terminology within the modeling community of intracellular reaction systems. In general statistical terminology, this task is called parameter estimation, and “model identification” sounds more like “model selection” which is generally considered to be a different problem, concerning choices about the structure of the model. For a general purpose scientific journal, this choice of terminology should be made clear.*

Authors’ response: Thanks for this comment. In the revised manuscript, we have changed the phrase “model identification” to “parameter identification” to avoid this ambiguity.

20. *Section 1E, fifth paragraph. Suggest: “~~In addition, it is noteworthy that [C]lassical particle filtering is generally ... For this reason, we require our identification algorithm to classify all of the parameters as follower components [(C4)]~~, which allows ...”*

Authors’ response: Thanks for this comment. In the revised manuscript, we have added “(C4)” after the phrase “follower components” to refer the readers to this specific condition. Please see the sentence right before C4 on Page 8. However, we chose to retain the phrase “it is noteworthy ...” because it underscores the significance of the following point.

21. *In the next paragraph, suggest: “~~It is also worth noting that [T]he idea of adapting the Rao-Blackwell method ...~~”*

Authors’ response: Please see the change in the second paragraph of the right column on Page 8.

22. *In paragraph 2 of section 2B, suggest: “This whole procedure took 10 graphics processing units (GPUs) [approximately] 24 hours[.]” (remove the exclamation mark). Similarly in section 2C, paragraph 2, remove the exclamation mark: “This procedure took 6.5 hours[.]”*

Authors’ response: In the revised manuscript, we have changed the punctuation as suggested.

23. *In section 2C, paragraph 3: “From figure 6E, we can observe that for the follower system, the solution of the RB-PF is more smooth and accurate [than the solution of the PF, demonstrating] the advantage of RB-PF.”*

Authors’ response: Thanks for this suggestion. We have revised this sentence accordingly. Please have a look at the first paragraph in the right column on Page 11.

24. *In section 3A, you say that the numeric simulations have a uniform prior, but if these are direct simulations from a distribution then they aren’t really prior distribution. Instead, you could say: “All these parameters were in units of minute^{-1} and were simulated from uniform distributions.”*

Authors’ response: Here, we mean all the parameters to have uniform prior distribution in the Bayesian estimation problem. We noticed that the original sentence “ k_1, \dots, k_4 were **drawn**

from ... ” is misleading. In the revised manuscript, we have rephrased it as “ k_1, \dots, k_4 **can take values** from ... ”. Please see the changes in the first paragraph of Section 3A on Page 12.

25. *The simulation study only used a single draw of parameters from a uniform distribution. Would the results be significantly different if more draws were taken? What if parameters came from distributions other than uniform?*

Authors’ response: Thanks for this comment related to the robustness of our method to the choice of actual parameter values. To address this question, we have added more simulation results in the revised manuscript, incorporating very different sets of actual parameter values. The result (presented in SI Appendix, section S7.C and Figure S1) shows that under different scenarios, our method can provide reliable estimation results, thereby demonstrating the effectiveness of our method. Please see these results in SI Appendix, section S7.C and Figure S1.

26. *Paragraph 3 of section 3A states that the conditional distributions of the parameters shown in Figure 9A have a high level of confidence. This seems to be an optimistic statement, as a significant amount of mass is considerably far away from the true parameter. For example, a 95% credible interval would probably cover the range [0.2, 0.7], which is half of the range of the considered values [0, 1].*

Authors’ response: In the revised manuscript, we have adjusted our language to state that these estimates are relatively confident. Please see the change in the third paragraph of the left column on Page 13.

27. *The manuscript has several footnotes, which should not be used: <https://www.nature.com/ncomms/submit/article>. One particular example is paragraph 4 of section 3A: “However, thanks to the stability is that the occupation time distribution ...”. This sentence doesn’t read very well, and I don’t see reason that the elaboration found in the footnote couldn’t be contained in the main text.*

Authors’ response: Thanks for this comment and pointing out the grammatical mistakes in the sentence. In the revised manuscript, we have removed all the footnotes by including all the information in the main text. We have also revised the sentence; please see the first paragraph in the right column on Page 13.

28. *In 3A there are two numeric simulations. In the second, where only two gene states are present in the target system, generating parameter values are provided in the main text; why are they explicitly provided here and not the first simulation?*

Authors’ response: In the first example, the parameter values were selected randomly, and as such, we did not specify them in the main text. However, these values can be identified in Fig. 9, as indicated by the red dashed lines. In the second numerical example, we intentionally set the values of k_1 , k_2 , and k_{p_1} to be the same as those in the first example, while setting the remaining parameters zero. This parameter choice is quite standard. It guarantees the system has only two effective gene states (as the third one can never be reached). Meanwhile, since the values of k_1 , k_2 , and k_{p_1} are chosen the same in both examples, the only difference in these examples is the number of gene states. Therefore, the success of our algorithm in both examples indicate that this difference doesn’t affect the effectiveness of our approach, and our approach can accurately identify whether a transcription system has two or three gene states.

In the revised manuscript, we have rephrased the sentence (introducing the parameters in the second example) by emphasizing that k_1 , k_2 , and k_{p_1} are identical to the selected values in the first example. Please see the change in the last paragraph of the right column on Page 13.

29. *Last paragraph of 3A you claim the stationary distributions of the target system and the inferred models match perfectly, but this isn’t true, they just match closely.*

Authors’ response: In the revised manuscript, we have changed the word as suggested.

30. *In Figure 10A you only show the results for cell #78. Why this particular cell? Obviously it would be difficult to show the results for all cells, but there should at least be some justification*

of why the results for this cell are representative of all of the cells. Do you get worse results if you look at others?

Authors' response: Thanks for this comment. The cell was randomly selected among the ones exhibiting a sufficiently active transcription process. We have emphasized this point in the revised manuscript; please see the last paragraph in the right column on Page 14. Our method showed consistent performance across all cells, as evidenced by the aggregated data in Figures 10B and 11B. As you correctly pointed out, it would be difficult to show the results for all cells in detail.

31. *In Figure 11 B: you say that all of the dots are distributed around the diagonal lines, but to me it seems that most points fall below the line, so the variance of the inferred model is consistently lower than the variance of the experiment, so the variance from the inferred models is consistently biased low.*

Authors' response: Thanks for pointing this out. We acknowledge that for many points, the variance of the inferred model is less than the variance estimated directly from experimental data via ergodicity, even though the dots are distributed around the diagonal line visually. By performing linear regression, the variance estimated by the inferred model is about 10% lower than the one estimated by the ergodicity property, which is not very significant. The result still suggests that the noise decomposition results obtained from the two methods are consistent. In the revised manuscript, we have pointed out this discrepancy in the caption of Fig 11; please see the caption on Page 18.

It is also important to consider that the benchmark variance estimated via ergodicity is not free from error. Due to the limited duration of the measurement period, the benchmark variance estimate has a bias. However, we have no explicit knowledge about whether this benchmark is underestimated or overestimated. Thus, we cannot judge whether the inferred model always provides underestimated variance. The discrepancy between the ergodicity-based estimates and those from the inferred model can also be attributed to the exclusion of certain biological mechanisms, such as cell cycles, photobleaching, etc. In the revised manuscript, we have mentioned these sources of discrepancies in the caption of Fig 11. Please see the caption of Fig 11 on Page 18

32. *The link between the Rao-Blackwell theorem and the so-called Rao-Blackwellized particle filters is not clear. This should have something to do with the relationship between the proposed algorithms and the Rao-Blackwellised particle filter of Kevin Murphy and Stuart Russell (see chapter 24 of Sequential Monte Carlo Methods in Practice, Arnaud Doucet et al).*

Authors' response: In the revised manuscript, we have added two references (including this one) related to the Rao-Blackwellized particle filter and cited them in the main text. Please see the references [74, 75], the second last paragraph in the left column on Page 3, and the second paragraph in the right column on Page 8.

33. *Several very similar algorithms are presented, with only a line or so different. If they could be combined into a single representation, with the few differing lines identified (e.g., formally by an if statement) that would save space and add clarity.*

Authors' response: Thanks for this suggestion. We have now combined the decomposition algorithms for the RB-CME solver and the RB-PF. Please see Algorithm 2 and Algorithm 3 in the supplementary information.

34. *In the supplement, line 81: "... can also solve the filtering problem by using the Monte Carlo for the CME step", should be "the Monte Carlo method", or otherwise remove the word "the".*

Authors' response: We have removed the word "the" in the revised manuscript.

35. *Supplement line 135: "we simplifies" should be "we simplify".*

Authors' response: We have corrected this typo in the revised manuscript. To prevent visual overload, we didn't highlight this change.

36. *Supplement line 194: "when discussing other conditional probability, we always refer to such a cadlag one". This sentence could be improved. Maybe something like: "when discussing other conditional probability, we are referring to a distribution that is cadlag".*

Authors' response: We have rewritten the sentence as suggested. To prevent visual overload, we didn't highlight this change.

37. *Supplement line 195: "we lists" should be "we list".*

Authors' response: Thanks. We have corrected this typo. To prevent visual overload, we didn't highlight this change.

38. *Supplement line 297: "we particular choose" could be "we specifically choose", or the word particular could be removed altogether.*

Authors' response: We removed the word "particular" in the revised manuscript.

39. *Supplement line 364, there is a typo, $h_j(\tilde{x}, z_1, \dots, z_1)$ should be $h_j(\tilde{x}, z_1, \dots, z_l)$.*

Authors' response: We have corrected it in the revised manuscript. To prevent visual overload, we didn't highlight this change.

40. *Supplement line 377, "We prove the rest result", remove the word "rest".*

Authors' response: We have remove the word "rest" in the revised manuscript.

41. *Supplement line 396: "Moreover, by math induction", formally, "Moreover, by the principle of mathematical induction,"*

Authors' response: We have rephrased it as suggested. To prevent visual overload, we didn't highlight this change.

42. *Supplement line 7 of Algorithm 6: The notation here could be improved. In particular, the samples $\tilde{z}_1(t_i), \dots, \tilde{z}_l(t_i)$ are unique for each particle. This is reflected for the "leader particles": $\tilde{x}_j(t_i)$, so this should also be j reflected in the follower particles, maybe: $\tilde{z}_{l,j}(t_i)$, or $\tilde{z}_l^j(t_i)$*

Authors' response: Thanks for this suggestion. We have replaced the notations with $\tilde{z}_1^j(t_i), \dots, \tilde{z}_l^j(t_i)$. Please see Line 7 of Algorithm 4.

43. *Supplement line 469: "A rigorous analysis for that requires the techniques in (4), which is somehow complicated.". Saying that it is somehow complicated feels a bit informal, maybe replace with "which could be the topic of future work", or "which is outside the scope of this analysis", or equivalent.*

Authors' response: In the revised manuscript, we stated that this rigorous analysis is left for future work. Please see Line 524 in the supplementary material.

References

- [BG01] Carlo Berzuini and Walter Gilks. Resample-move filtering with cross-model jumps. In *Sequential Monte Carlo Methods in Practice*, pages 117–138. Springer, 2001.
- [CGP05] Yang Cao, Daniel T Gillespie, and Linda R Petzold. The slow-scale stochastic simulation algorithm. *The Journal of chemical physics*, 122(1):014116, 2005.
- [CM18] DAN CRISAN and JOAQUÍN MÍGUEZ. Nested particle filters for online parameter estimation in discrete-time state-space markov models. *Bernoulli*, 24(4A):3039–3086, 2018.
- [EVE05] Liu D EW and E Vanden-Eijnden. Nested stochastic simulation algorithm for chemical kinetic systems with disparate rates. *J Chem Phys*, 123(19):194107, 2005.
- [FGK23a] Zhou Fang, Ankit Gupta, and Mustafa Khammash. Convergence of regularized particle filters for stochastic reaction networks. *SIAM Journal on Numerical Analysis*, 61(2):399–430, 2023.
- [FGK23b] Zhou Fang, Ankit Gupta, and Mustafa Khammash. Effective filtering approach for joint parameter-state estimation in sdes via rao-blackwellization and modularization. *arXiv preprint arXiv:2311.00836*, 2023.

- [GB01] Walter R Gilks and Carlo Berzuini. Following a moving target—monte carlo inference for dynamic bayesian models. *Journal of the Royal Statistical Society: Series B (Statistical Methodology)*, 63(1):127–146, 2001.
- [Gou05] John Goutsias. Quasiequilibrium approximation of fast reaction kinetics in stochastic biochemical systems. *The Journal of chemical physics*, 122(18):184102, 2005.
- [KK13] Hye-Won Kang and Thomas G Kurtz. Separation of time-scales and model reduction for stochastic reaction networks. *The Annals of Applied Probability*, 23(2):529–583, 2013.
- [KS17] Jae Kyoung Kim and Eduardo D Sontag. Reduction of multiscale stochastic biochemical reaction networks using exact moment derivation. *PLoS computational biology*, 13(6):e1005571, 2017.
- [LW01] Jane Liu and Mike West. Combined parameter and state estimation in simulation-based filtering. In *Sequential Monte Carlo methods in practice*, pages 197–223. Springer, 2001.
- [OM00] Nadia Oudjane and Christian Musso. Progressive correction for regularized particle filters. In *Proceedings of the Third International Conference on Information Fusion*, volume 2, pages THB2–10. IEEE, 2000.
- [PVM21] Sara Pérez-Vieites and Joaquín Míguez. Nested gaussian filters for recursive bayesian inference and nonlinear tracking in state space models. *Signal Processing*, 189:108295, 2021.
- [SHK21] Yun Min Song, Hyukpyo Hong, and Jae Kyoung Kim. Universally valid reduction of multiscale stochastic biochemical systems using simple non-elementary propensities. *PLoS Computational Biology*, 17(10):e1008952, 2021.
- [ZUP⁺14] Christoph Zechner, Michael Unger, Serge Pelet, Matthias Peter, and Heinz Koepl. Scalable inference of heterogeneous reaction kinetics from pooled single-cell recordings. *Nature methods*, 11(2):197–202, 2014.

Reviewer #1 (Remarks to the Author):

The authors have successfully responded to comments.

Reviewer #2 (Remarks to the Author):

See attached report

Reviewer #2 (Remarks on code availability):

I did not install and run the code. I checked it is available at this location. From a cursory read, it appears to be written and documented in compliance with expectations for scientific computing.

Reviewer #2 Attachment on the following page

Review of the revised manuscript: Advanced methods for gene network identification and noise decomposition from single-cell data

March 19, 2024

The revised manuscript addresses most of our previous concerns. We have some comments related to the revisions.

1 Main Points

1. Based on a recommendation of Reviewer 1, the revised manuscript considers the estimation of the noise parameter σ in the numerical case studies (section 2). This addition has notably strengthened the manuscript, as accurately determining noise parameters is crucial for inference. However, in the analysis of experimental data (3.B.1), the value of σ is still assumed to be 1, despite explicitly noting that “the specific magnitude of the measurement noise has not been exactly quantified in literature”. Given that it is possible to estimate this parameter, and the value is unknown, we suggest it would be beneficial to also estimate σ for the experimental data, just as was successfully conducted in the numerical case study.
2. The revised manuscript includes an additional sentence that was intended to address a previous concern about the need to discretize the parameter space:

“It can also be attributed to the coarse discretization scheme in the parameter space. We have also proposed an improved RB-PF based on grid refinement, which can reduce this discrepancy by one-third.”

If the authors choose to extend the analysis in the main paper to include estimation of σ , they may want to use their improved RB-PF in the main article, to emphasize its capabilities.

3. This new grid refinement algorithm seems to involve using the posterior obtained from a Bayesian analysis (in one refinement stage) to inform a prior distribution that will later be used to perform a Bayesian analysis at the next stage. This re-use of data is sometimes called “double-counting” or “double-dipping”, and should be noted, since the problem solved may no longer exactly match the original Bayesian prior and posterior.
4. The newly added results in the supplementary material confirm that a major limitation of the proposed algorithm is that it requires a discretized parameter space. Figure 10A shows a high level of confidence for k_{p1} being 32 versus the other nearest possible values of 24 or 40. On the finer scale in Figure S3, however, evidence of bi-modality emerges, with the largest mode occurring at the value $k_{p1} = 30$, demonstrating distinct dynamics at $k_{p1} = 30$ vs $k_{p1} = 32$; the difference in dynamics is also evidenced by the fact that finer scale reduced the discrepancy in stationary distributions by one third when using a finer scale. The risks and benefits of discretization may be apparent to readers, but some additional clarification would help make sure of this.
5. On a similar topic, we don’t fully agree with a comment you made in your response:

However, the discretization sizes (8 and 10) are actually not very large compared to the parameter intervals $[0, 80]$ and $[20, 120]$, and they are sufficient to allow for accurate estimates of the parameters (evident by results we presented in that section).

The step size relative to the possible parameter interval is not the proper quantity to consider. Instead, what matters is the range that can be effectively distinguished using observed data. It’s possible that the set of plausible values may be large (say $[0, 80]$), yet there is a very strong signal in the data that makes the region of high-probability for a particular variable to be very small (say, 30 ± 0.5). Unless at least one of the discrete points considered falls in this region of high probability (which cannot be known a priori), your estimates will be biased.

2 Minor Points

1. After Eq [2], I believe that the description for ζ_j is incorrect. It should be:

$$(\nu'_{1,j} - \nu_{1,j}, \dots, \nu'_{n,j} - \nu_{n,j})^T?$$

2. “By classifying the chemical species differently, one can choose whether this method is similar to the Monte-Carlo method or to the chosen filtering approach.” This sentence should be re-worded. Your method—other than in the extreme cases with all leader-level or no-leader level species—always has similarities to both MC and the chosen filtering approach; the current wording suggests the similarity may only exist depending on the choice of classifying the chemical species.
3. In the last paragraph of section 1D: “usually has similar accuracy to the RB-CME solver”. We would recommend against the use of the word “usually” here, as it makes an unclear general claim. You have provided a reference following this statement to a mathematical analysis performed in the supplement, suggesting a better wording may be “... the RB-PF (for the filtering problem) as similar accuracy to the RB-CME solver (for solving CMEs) under certain regularity conditions (SI Appendix, Section S4.C), ...”. Alternatively, the statement “usually” may be referring to your numeric results, in which case it would be worth specifying: “the RB-PF (for the filtering problem) has similar accuracy to the RB-CME solver (for solving CMEs) in all of the case studies considered in this article”.
4. **Minor suggestion:** “This procedure took 6.5 hours”. Statements of computational time without details of the number of cores / the amount of parallelization done in the calculation are not very helpful. If this was on a single core, please be explicit about that; otherwise, give the number of cores used.
5. **Minor suggestion:** “also, we need to point out”. The need to point this out is already evident in the fact that you are drawing attention to it. Simply stating: “The method in (11) is not applicable to this system, as it requires the system to have distinct reaction vectors ...” (or similar) is sufficient.
6. **Minor suggestion:** In Eq. [7], σ is a diagonal matrix. Would not Σ be a more standard choice for this matrix? The symbol σ also coincides with a different parameter that was previously used.
7. **Minor suggestion:** “..., which is not trivial to a priori determine for generic systems”. While this statement is not grammatically incorrect, it is more common to reverse the statement: “..., which is not trivial to determine a priori for generic systems”.
8. **Minor suggestion:** the emphasis on the word “several” in the abstract is not needed and may be distracting.

Reviewer #3 (Remarks to the Author):

Authors' response to the reviewers and list of actions taken

Zhou Fang, Ankit Gupta, Sant Kumar, Mustafa Khammash

April 15, 2024

We thank the referees for the positive recommendation of our manuscript. Responding to the referees' comments has helped us considerably improve the quality of the manuscript. The major changes in this revision were:

- Addition of inference results for estimating the observation noise intensity σ from the experimental data to further illustrate the effectiveness of our approach.
- More discussion about how to properly utilize the grid refinement strategy.

We have addressed all of the issues raised by the referees in the revised manuscript. Below, we list the referees' comments, followed by our detailed response and the actions taken. For the convenience of the editor and the referees, we have highlighted in blue the parts of the manuscript that have been changed.

Response to Referee 2 and 3

1. *Based on a recommendation of Reviewer 1, the revised manuscript considers the estimation of the noise parameter σ in the numerical case studies (section 2). This addition has notably strengthened the manuscript, as accurately determining noise parameters is crucial for inference. However, in the analysis of experimental data (3.B.1), the value of σ is still assumed to be 1, despite explicitly noting that "the specific magnitude of the measurement noise has not been exactly quantified in literature". Given that it is possible to estimate this parameter, and the value is unknown, we suggest it would be beneficial to also estimate σ for the experimental data, just as was successfully conducted in the numerical case study.*

Authors' response: In the revised manuscript, we have added an inference result for the observation noise intensity σ from the experimental data. The result shows that our method can confidently estimate the parameter σ to be 0.4, confirming the accuracy of the mRNA measurements. Also, this result indicates that the actual observation noise intensity is much lower than what we expected (i.e., $\sigma = 1$). However, this more accurate estimation of σ does not significantly impact the inference results for the other parameters and, therefore, does not contribute to further reduction of the discrepancy between the inferred model and the actual system. More details can be found on Page 16 (left column), Section S7.F in the supplementary material, and Figure S4 in the supplementary material.

2. *The revised manuscript includes an additional sentence that was intended to address a previous concern about the need to discretize the parameter space:*

"It can also be attributed to the coarse discretization scheme in the parameter space. We have also proposed an improved RB-PF based on grid refinement, which can reduce this discrepancy by one-third."

If the authors choose to extend the analysis in the main paper to include estimation of σ , they may want to use their improved RB-PF in the main article, to emphasize its capabilities.

Authors' response: Thanks for this suggestion. We have applied the improved RB-PF for estimating σ . Our algorithm confidently estimated σ to be 0.4, suggesting that the mRNA measurement is even more accurate than what we previously assumed (i.e., $\sigma = 1$). However, this more accurate estimate of parameter σ does not significantly impact the inference results

for the other parameters (see the details Section S7.F in the supplementary material, and Figure S4 in the supplementary material). Therefore, we still keep the original results with σ assumed to be 1 in the main text and put the newly added inference results (which include the estimation of σ) in the supplementary material.

3. *This new grid refinement algorithm seems to involve using the posterior obtained from a Bayesian analysis (in one refinement stage) to inform a prior distribution that will later be used to perform a Bayesian analysis at the next stage. This re-use of data is sometimes called “double-counting” or “double-dipping” and should be noted since the problem solved may no longer exactly match the original Bayesian prior and posterior.*

Authors’ response: In the revised supplementary material, we have added a paragraph to discuss the potential issue related to “double-dipping” and inform the readers that this refinement strategy should be used with care. Please see the revision in Section S7.E.

In our case, since we used the uniform prior distributions, we can show that we still recover the correct Likelihoods at each inference stage, provided that we only discard parameter regions with very small likelihoods in the previous stage. It is explained in greater detail in Section S7.E.

4. *The newly added results in the supplementary material confirm that a major limitation of the proposed algorithm is that it requires a discretized parameter space. Figure 10A shows a high level of confidence for k_{p_1} being 32 versus the other nearest possible values of 24 or 40. On the finer scale in Figure S3, however, evidence of bi-modality emerges, with the largest mode occurring at the value $k_{p_1} = 30$, demonstrating distinct dynamics at $k_{p_1} = 30$ vs $k_{p_1} = 32$; the difference in dynamics is also evidenced by the fact that finer scale reduced the discrepancy in stationary distributions by one third when using a finer scale. The risks and benefits of discretization may be apparent to readers, but some additional clarification would help make sure of this.*

Authors’ response: In the revised supplementary material, we added a paragraph to discuss the risks of using a coarse grid. Also, we inform the readers to use the grid refinement strategy whenever a large region of low posterior is observed. Please see the discussion in the last paragraph of Section S7.E in the supplementary material.

5. *On a similar topic, we don’t fully agree with a comment you made in your response:*

However, the discretization sizes (8 and 10) are actually not very large compared to the parameter intervals [0, 80] and [20, 120], and they are sufficient to allow for accurate estimates of the parameters (evident by results we presented in that section).

The step size relative to the possible parameter interval is not the proper quantity to consider. Instead, what matters is the range that can be effectively distinguished using observed data. It’s possible that the set of plausible values may be large (say [0, 80]), yet there is a very strong signal in the data that makes the region of high-probability for a particular variable to be very small (say, 30 ± 0.5). Unless at least one of the discrete points considered falls in this region of high probability (which cannot be known a priori), your estimates will be biased.

Authors’ response: We acknowledge that the resolution should not be only evaluated by the ratio between the step size and the parameter interval. Instead, it should be evaluated whether the peaks and valleys of the conditional distributions are able to be identified. In the revised supplementary, we have added a sentence to explain this point. Please see the last sentence in the last paragraph of Section S7.E in the supplementary material.

6. *After Eq [2], I believe that the description for ζ_j is incorrect. It should be: $(\nu'_{1,j} - \nu_{1,j}, \dots, \nu'_{n,j} - \nu_{n,j})$?*

Authors’ response: Thanks for this comment. We have corrected this typo in the revised manuscript. Please see the change below Eq. [2].

7. *“By classifying the chemical species differently, one can choose whether this method is similar to the Monte-Carlo method or to the chosen filtering approach.” This sentence should be reworded. Your method—other than in the extreme cases with all leader-level or no-leader level species—always has similarities to both MC and the chosen filtering approach; the current*

wording suggests the similarity may only exist depending on the choice of classifying the chemical species.

Authors' response: In the revised manuscript, we have clarified that regardless of how the chemical species are classified, our method consistently integrates both the Monte-Carlo method and the chosen filtering approach. We rephrased the sentence as "In the scenarios between these extreme cases, our method is a combination of the two approaches, capable of leveraging their advantages to achieve better performance than either method in isolation." Please see the change on Page 6 (left column).

8. *In the last paragraph of section 1D: "usually has similar accuracy to the RB-CME solver." We would recommend against the use of the word "usually" here, as it makes an unclear general claim. You have provided a reference following this statement to a mathematical analysis performed in the supplement, suggesting a better wording may be "... the RB-PF (for the filtering problem) as similar accuracy to the RB-CME solver (for solving CMEs) under certain regularity conditions (SI Appendix, Section S4.C), ...". Alternatively, the statement "usually" may be referring to your numeric results, in which case it would be worth specifying: "the RB-PF (for the filtering problem) has similar accuracy to the RB-CME solver (for solving CMEs) in all of the case studies considered in this article."*

Authors' response: Thanks for this suggestion. We have revised the sentence accordingly. Please see this change in the last paragraph of Section 1.D.

9. *"This procedure took 6.5 hours". Statements of computational time without details of the number of cores / the amount of parallelization done in the calculation are not very helpful. If this was on a single core, please be explicit about that; otherwise, give the number of cores used.*

Authors' response: In the original version of our paper, we have mentioned the computational setup at the beginning of Section 2. Unless otherwise stated, all the experiments were performed on the Euler computing cluster at ETH Zurich, utilizing computational nodes with 2.25-GHz and 12-core CPUs. Therefore, the reported computational time of 6.5 hours for the mentioned procedure was achieved using a 12-core CPU. To make this point clear, we have revised the sentence as "This procedure took 6.5 hours on a 12-core CPU." Please see the change on Page 11 (left column).

10. *"Also, we need to point out". The need to point this out is already evident in the fact that you are drawing attention to it. Simply stating: "The method in (11) is not applicable to this system, as it requires the system to have distinct reaction vectors ..." (or similar) is sufficient.*

Authors' response: We have revised the sentence as suggested. Please see the change on Page 12 (left column).

11. *In Eq. [7], σ is a diagonal matrix. Would not Σ be a more standard choice for this matrix? The symbol σ also coincides with a different parameter that was previously used.*

Authors' response: We have revised Eq. [7] as suggested. Please refer to Eq. [7] on Page 7. In the paper, when the observation process is one dimensional and the matrix Σ is therefore one-by-one, we still use σ to represent the intensity of the only observation noise, because this is a more natural notation in this context. This point is mentioned in the text following Eq. [7]. Despite a thorough review, we could not identify any use of the symbol σ for parameters other than the observation noise intensity. Therefore, we believe using σ to represent the observation noise intensity, when only one observation channel is present, is fine.

12. *"..., which is not trivial to a priori determine for generic systems". While this statement is not grammatically incorrect, it is more common to reverse the statement: "..., which is not trivial to determine a priori for generic systems".*

Authors' response: We have revised it as suggested. Please see this change in the second paragraph of the right column on Page 2.

13. *The emphasis on the word "several" in the abstract is not needed and may be distracting.*

Authors' response: We have removed the emphasis on the word "several" in the abstract.

Reviewer #2 (Remarks to the Author):

The authors have attended to all my comments. In my opinion, their manuscript is ready for publication.

Reviewer #2 (Remarks on code availability):

On this round of review, I checked only to see how adequately my previous comments from earlier rounds were addressed.

Reviewer #3 (Remarks to the Author):

Thank you for your response and revisions, the authors have successfully responded to my comments and concerns.
